# Human induced pluripotent stem cell-derived vocal fold mucosa mimics development and responses to smoke exposure

Vlasta Lungova [1], Xia Chen [1], Ziyue Wang [2], Christina Kendziorski[2] & Susan L. Thibeault [1]

Development of treatments for vocal dysphonia has been inhibited by lack of human vocal fold (VF) mucosa models because of difficulty in procuring VF epithelial cells, epithelial cells' limited proliferative capacity and absence of cell lines. Here we report development of engineered VF mucosae from hiPSC, transfected via TALEN constructs for green fluorescent protein, that mimic development of VF epithelial cells in utero. Modulation of FGF signaling achieves stratified squamous epithelium from definitive and anterior foregut derived cultures. Robust culturing of these cells on collagen-fibroblast constructs produces three-dimensional models comparable to in vivo VF mucosa. Furthermore, we demonstrate mucosal inflammation upon exposure of these constructs to 5% cigarette smoke extract. Upregulation of pro-inflammatory genes in epithelium and fibroblasts leads to aberrant VF mucosa remodeling. Collectively, our results demonstrate that hiPSC-derived VF mucosa is a versatile tool for future investigation of genetic and molecular mechanisms underlying epithelium-fibroblasts interactions in health and disease.

---

[1] Department of Surgery, University of Wisconsin Madison, Wisconsin Institute for Medical Research, Madison, WI, USA. [2] Department of Biostatistics & Medical Informatics, University of Wisconsin Madison, Madison, WI, USA. Correspondence and requests for materials should be addressed to S.L.T. (email: thibeault@surgery.wisc.edu)

Human voice is vital to our social being and quality of life. Impaired voice production holds significant implications for individual health and wellness, occupational function, and societal productivity. The basis or foundation to this acoustic communication are the vocal folds (VF) that are housed in the larynx, at the crossroads between the respiratory and digestive tracts. Besides mechanical stresses that occur during phonation, VF are exposed to environmental insults, allergens, smoke, reflux, and viral, bacterial, or fungal infections. Exposure to irritants over time may lead to chronic laryngeal inflammation that can cause VF injuries both benign, such as nodules[1,2], polyps[1,2], granulomas[3,4], and malignancy[5,6]. Besides chronic inflammation, smoking is a major risk factor of laryngeal keratosis, Reinke's edema, and laryngeal leukoplakia[7–9]. Treating these disorders is associated with substantial and far-ranging social, psychological, and economic costs exceeding $11 billion per year[1,2,10–12] similar to asthma, heart disease, and depression[11,12].

VF stratified squamous epithelium is a multifunctional interface that is necessary for maintaining VF homeostasis. Although dysfunction of the epithelium may play a significant pathogenic role in VF diseases, studies at the genetic and molecular level using primary VF epithelial cells (VFEC) or models of human VF mucosa have been significantly limited by the availability of relevant tissue types derived from biopsies or cadaveric sources, poor growth, heterogeneity of primary VFEC, and absence of VFEC lines[13,14]. Primary VFEC cannot be removed from a healthy larynx without a significant risk to VF function, as compared with other epithelial surfaces, such as nasal and airway epithelium[15–19]. Absence of a reliable in vitro model of human VF mucosa for studying pathological changes in voice inhibits the development of drugs or treatments that could target aberrant VF repair in a disease and/or patient-specific manner.

Recent developments in stem cell biology promise cell sources for modeling genetic or environmental diseases in VF with organotypic in vitro culture systems. Besides serving as three-dimensional (3D) models for clinical and pharmacological applications, these constructs can be potentially utilized for formation of bioengineered VF for organ replacement[20]. The main goals of this study were to generate reproducible hiPSC-derived VF epithelium that mimics development of VF epithelium in utero and create in vitro human VF mucosa composed of hiPSC-derived stratified squamous VFEC and collagen gel seeded with human primary VF fibroblasts (VFF). Functional validation of these constructs was performed by exposing engineered VF mucosae to 5% cigarette smoke extract (CSE) for 1 week to mimic cigarette smoke exposure. In order to distinguish hiPSC-derived VFEC from primary VFF, we transfected hiPSC with green fluorescent protein (GFP) prior to their differentiation into the human AAVS1 locus[21]. A stably integrated fluorescent gene into human iPS cells enabled us to assess morphological and functional changes in response to the cigarette smoke extract in both cell types, simultaneously.

In this study, we establish a protocol for derivation of the VF epithelium based upon our previous work in a mouse model that demonstrated that VF stratified squamous epithelial cells are derived from definitive endoderm (DE) and anterior foregut endoderm (AFE) similar to other respiratory surfaces and esophagus[22–24]. HiPSC were first differentiated into DE and AFE[25–27], then AFE cells were exposed to high levels of fibroblast growth factor (FGF) to induce stratification and differentiate into vocal fold basal progenitors (VBP) which were reseeded on top of collagen–fibroblast constructs and cultivated, first as submerged cultures in the presence of high FGF and then at an air–liquid interface (A/LI). Before plating on collagen–fibroblast constructs, the transcriptional profile of hiPSC-derived VBP was compared with human primary VFEC and human fetal esophagus[24] using

RNA sequencing to confirm VF-specific differentiation. After 32 days in culture, hiPSC-derived VF mucosae were characterized by immunohistochemistry (IHC) and quantitative polymerase chain reaction (qPCR). Derived VFEC demonstrated key morphologic, genotypic, and phenotypic similarities to native VF epithelium.

To examine whether this system has capability to address clinically relevant problems in laryngology, we exposed the engineered VF mucosa to 5% cigarette smoke extract (CSE) for 1 week to mimic cigarette smoke exposure. Detrimental effects of smoking on VF health have been confirmed both clinically and experimentally[28–30]. Smoking is a major risk factor of chronic laryngeal inflammation that can consequently lead to keratotic changes in VF mucosa[30,31]. In this study, we show that exposure of engineered VF mucosa to 5% CSE for 1 week induced morphological and functional changes in VFEC and VFF that lead to aberrant VF mucosa remodeling. These changes include altered morphology of basal and luminal cellular compartments, and impaired mucus production. Morphological and functional changes in VFEC may be triggered by the activation of pro-inflammatory genes detected in VFEC as well as VFF in the collagen matrix. Collectively, our results demonstrate that our hiPSC-derived VF mucosa supports a spectrum of physiological functions and possesses relevant phenotypic plasticity to probe important features of voice pathologies. This versatile model system can be used to elucidate molecular mechanisms underlying epithelium–fibroblasts interactions in both health and disease, and could be utilized to study aberrant VF remodeling, including identification of targets for possible genetic and/or pharmacological manipulations.

## Results

**Generation of GFP reporter hiPSC line**. In order to study responses of hiPSC-derived VFEC and human primary VFF in engineered VF mucosae separately, we labeled hiPSC with GFP prior to their differentiation. HiPSC (IMR-90-4) were transfected with GFP by targeting a GFP reporter gene with puromycin-resistance cassette to the human AAVS1 locus. We performed homologous recombination using commercially available TALEN constructs[21]. After successful transfection with specific AAVS1-Talens and AAV-CAGGS-EGFP donor plasmids and drug selection (Fig. 1a), a total of three clones were expanded and screened by PCR and southern blotting for homologous recombination events[21]. For PCR screening, we amplified the junction between the AAVS1 genome and inserted donor. In GFP-positive clones, 5′-junction PCR primers produced a 1033 bp product (Fig. 1b). In southern blot analysis, the internal probe was located inside the left homology arm AAVS1-LA (705 bp). An uncropped blot is supplied in Source data, Supplementary Fig. 4. It recognized targeted GFP integration with SphI enzyme digestion[21]. Our results show a successful GFP integration (band size 3.8 kbp) (Fig. 1b)[21]. Primers for PCR and southern blot (Supplementary Table 1) were designed[21]. Clone B10 was selected for further differentiation; GFP transfection was persistent and robust in long-term cell cultures (Fig. 1c–n).

**Sequential differentiation of hiPSC into VF basal progenitors**. We developed a protocol for directed differentiation of hiPSC into VBP that could be reseeded on the top of collagen constructs and stratified at an A/LI. We used endodermal cell lineage differentiation through DE and AFE to mimic development of VBP in utero[22]. To derive VBP, hiPSC were first differentiated into DE for 3 days and then into AFE for 4 days. We induced human DE formation using Activin A (ACTA) (100 ng/ml) for 3 days supplemented with Wnt3a (25 ng/ml) day 1, and 0.2% fetal bovine

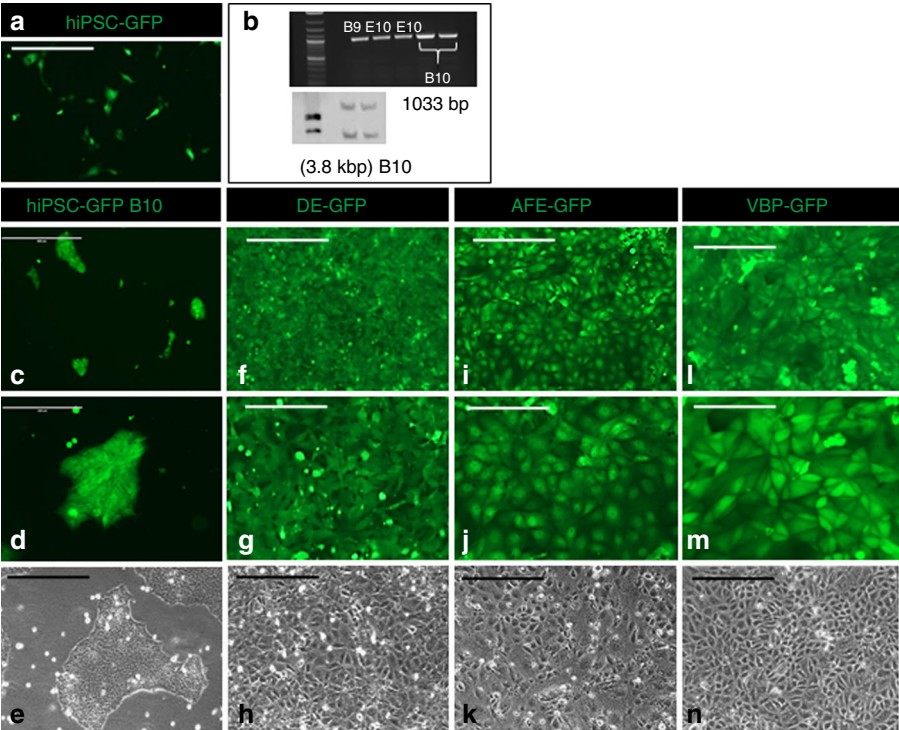

**Fig. 1** Green fluorescent protein incorporation into the genome of hiPSC. **a** Green GFP-positive colonies after nucleofection. **b** Confirmation of correct GFP insertion with the polymerase chain reaction and southern blotting. **c–e** Colonies of green GFP-positive B10 clones selected for differentiation (**c**). A detail of the B10 GFP + colony (**d**). A bright-field image of B10 GFP + colony (**e**). **f–n** show stable and persistent GFP insertion throughout cell differentiation, including bright field images, in the DE (**f–h**), AFE (**i–k**), and VBP before seeding on the collagen constructs (**l–n**). Scale bars in the panels of **a**, **c**, **e**, **f**, **h**, **i**, **k**, **l**, **n** = 400 μm. Scale bars in the panels of **d**, **g**, **j**, and **m** = 200 μm. AFE anterior foregut endoderm, DE definitive endoderm, GFP green fluorescent protein, hiPSC human induced pluripotent stem cells, VBP vocal fold basal progenitors. Source data provided as a Source Data file

serum (FBS) for an additional 2 days in the RPMI medium[25,32,33]. We treated cells with Noggin 200 ng/ml and SB 431542 (10 μM) for an additional 4 days in DMEM/F12 medium supplemented with B27 and N2, ascorbic acid, and monothioglycerol (called DMEM basal medium) (Fig. 2a) to induce AFE formation[25,26,34]. Derivation of DE and AFE was verified by immunocytochemistry (ICC), qPCR, and flow cytometry. ICC and qPCR methods confirmed that DE conditions were able to induce SOX17 and FOXA2 expression, while reducing SOX2 (Fig. 2b–d). DE cells expressed EPCAM and initiated expression of a simple epithelial marker cytokeratin (K) 8 (Fig. 2e, f). In AFE, mRNA and protein expression levels of SOX2 were re-established, FOXA2, endodermal marker, was maintained, while repressing DE marker SOX17 and intestinal lineage marker CDX2 (Fig. 2b, h, i)[25,26,32–34]. AFE cells expressed a basal cell marker p63, and K8, however, expression levels of stratified markers such as K5, K14, or K13 were low (Fig. 2g, j, k).

Simultaneously, we subjected DE and AFE cell populations to sorting to confirm presence of GFP-, DE-, and AFE-specific surface markers (Fig. 2l, m). For DE cell populations, we obtained 98.5 ± 0.8% of cells GFP-positive. Gating for DE surface markers CXCR4+ and EPCAM + generated 89.6 ± 5.6% double-positive cells (Fig. 2l), confirming that IMR-90-4 cell line was suitable for DE and AFE derivation[34]. For AFE, flow cytometry confirmed GFP (99.4 ± 0.8% positive) and SOX2 basal epithelial cell markers NGFR + and CD56 + . Sorting gates were established[25]. We were able to generate 89.3 ± 16.7% double-positive cells (Fig. 2m), suggesting that AFE cells can simultaneously represent VBP that can stratify when seeded on collagen gel constructs. To test this, we trypsinized AFE cells and plated them on top of collagen constructs with human primary VFF in flavinoid adenine

dinucleotide medium (FAD)[35]. Direct reseeding of AFE cells on collagen matrix in FAD medium did not lead to proper cell attachment, suggesting that AFE-derived cells were still immature. We determined that inducing stratification is necessary to improve attachment of AFE cells to collagen and promote their further differentiation at the A/LI interface.

We further focused on defining conditions that could stimulate expression of stratified markers in AFE cells to derive VBP. Our previous investigation in the mouse demonstrated that sonic hedgehog (SHH) signaling is important for early stages of VF development[22]. Ablation of SHH in mice foregut endoderm leads to serious defects in VF epithelium, disrupting SOX2 and NKX2-1 expression, resulting in laryngeal and VF agenesis[22]. SHH signals from the endoderm to mesenchyme; mutations in the GLI3 gene in *Gli3−/−* mutant mice leads to aberrant accumulation of thyroglottal tissue affecting insertion of the vocalis muscle and vocalization[36].

Based on mouse studies, we hypothesized that increased expression of SHH signaling may act on differentiation of VF basal cells and induce stratification in AFE-derived cells. Previous work demonstrated that high levels of FGF can stimulate SHH expression in human foregut endoderm cultures to derive lung organoids[27]. Given that the larynx and VF develop in the same ventral domain of the anterior foregut, we first tested whether exposure of AFE cells to FGF can stimulate SHH expression and expression of stratified markers in VBP. We exposed AFE-derived cells for 4 additional days to four different experimental conditions. At day 8, AFE cell cultures were treated with plain DMEM basal medium (group 1), then with DMEM basal medium supplemented with a low FGF2 concentration (50 ng/ml) (group 2), high FGF2 concentration (250 ng/ml) (group 3),

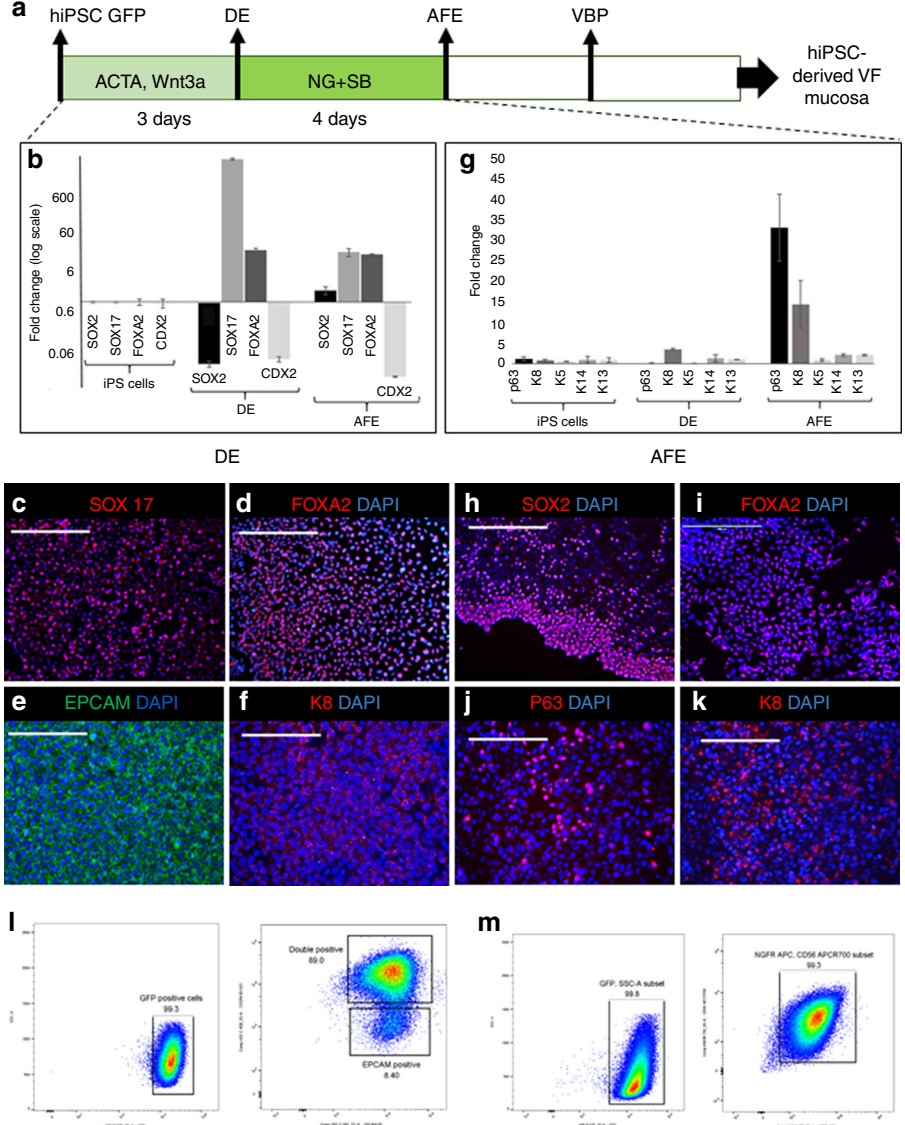

**Fig. 2** Generation of anterior foregut cell cultures from DE monolayers. **a** Differentiation of hiPSC into the DE and AFE. **b–k** Expression levels of the markers of the DE and AFE during differentiation characterized with quantitative polymerase chain reaction (**b**, **g**) and immunofluorescent staining for SOX17 (**c**), FOXA2 (**d**, **i**), EPCAM (**e**), cytokeratin K8 (**f**, **k**), SOX2 (**h**), and p63 (**j**), scale bar = 200 μm. Controls in **b**, **g** are hiPSC. In **b**, value of 0 = 1 for hiPSC control, and in **g** 0 = 0 expression for controls. We used one-way ANOVA to assess statistical significance between gene expression levels. Except for cytokeratin K13 with $p$-value = 0.06, $p$-values for other tested genes were below the threshold $p$-value < 0.05 indicating that expression levels of these genes significantly differed during differentiation. Error bars represent ± standard error of the mean, $n = 3$ per group. **l**, **m** Gating strategy for confirmation of DE and AFE cell populations by flow cytometry. Cell populations were sorted based on GFP expression and CXCR4 and EPCAM for DE (**l**) and CD56 and NGFR + for AFE (**m**). ACTA activin A, AFE anterior foregut endoderm, DE definitive endoderm, hiPSC human induced pluripotent stem cells, VBP vocal fold basal progenitors. Source data provided as a Source Data file

and combination of high levels of FGF2 (250 ng/ml), FGF10 (100 ng/ml), and FGF7 (100 ng/ml) (group 4) with the medium changed every other day (Fig. 3a). At day 12, we compared the transcript levels of SHH and stratification markers by qPCR. All experimental groups were compared with AFE controls collected at day 8 of differentiation. We found that SHH expression levels correlate with increases in FGF expression and are dose dependent (Fig. 3b). Exposure of AFE cells to the highest concentration of FGF (group 4) increased SHH expression in VBP as compared with cell cultures exposed to plain DMEM basal medium (group 1), and low or high levels of FGF2 only (Fig. 3b). Our results indicate that gradually increased levels in SHH signaling does not correlate with upregulation of stratified markers such as p63, K5, K14, and K13, suggesting that

upregulation of stratified markers is independent on SHH signaling (Fig. 3b). Only AFE cells exposed to the mixture of FGF signals, FGF2, FGF10, and FGF7 (group 4), were able to induce an expression of K5, K14, and K13 and simultaneously preserve high p63 and lower K8 expression (Fig. 3b). These results indicate that high levels of FGF signaling, particularly FGF10 and FGF7, can be involved in inducing stratification of AFE in cell cultures to mimic the transition between simple to stratified VF basal progenitors documented in vivo.

To qualitatively assess hiPSC-derived VBP, we conducted RNA sequencing (RNA-seq) to compared transcriptional profiles of hiPSC-derived VBP treated with the mixture of FGF for 4 days to human adult primary VFEC and human fetal esophageal epithelia[24] to confirm VF-specific differentiation. We performed

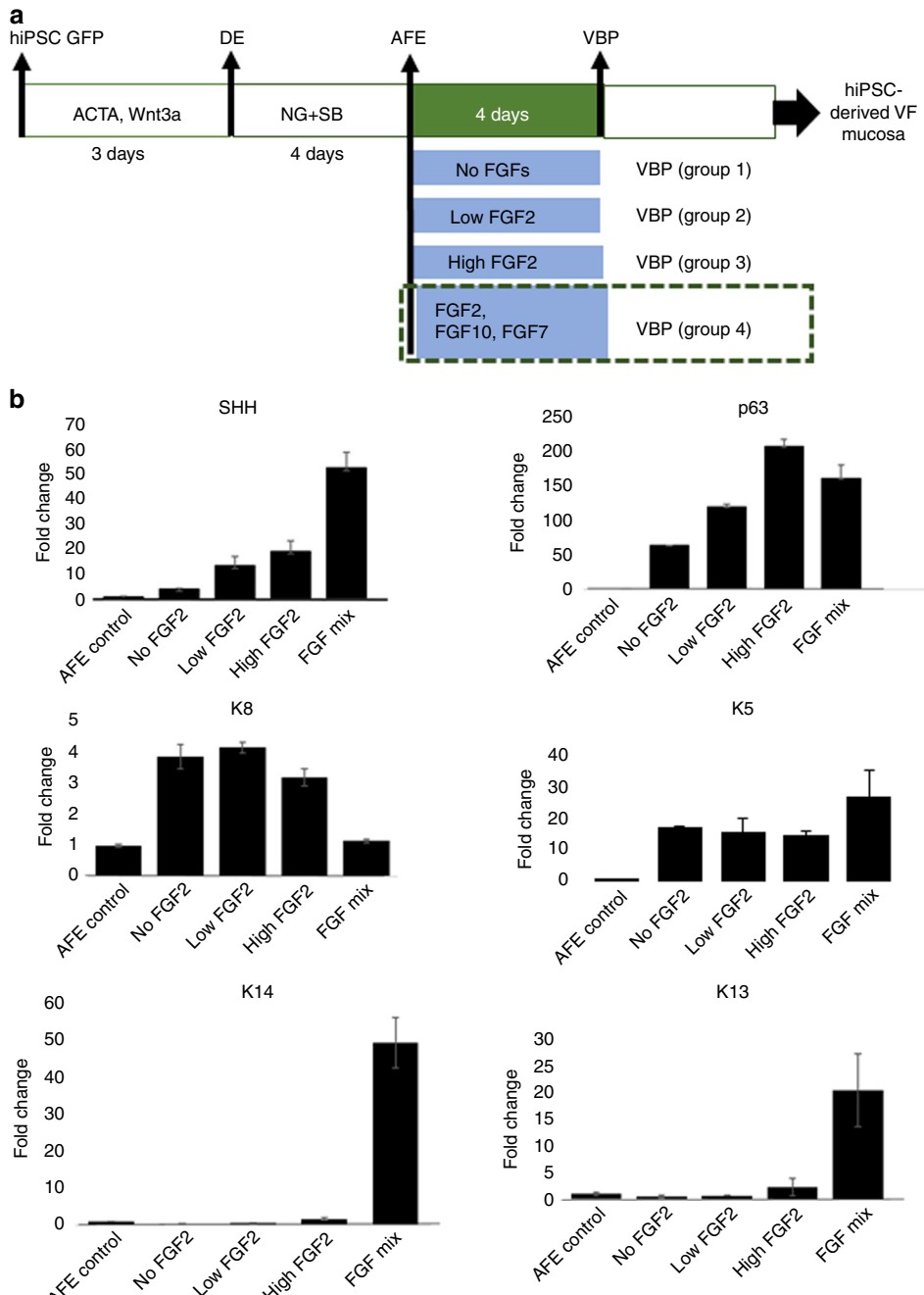

**Fig. 3** Induction of stratification in the AFE by modulating FGF signaling. **a** hiPSC were differentiated into the DE and AFE. AFE-derived cell cultures were then treated with various concentrations of fibroblasts growth factors (FGF2, FGF7, and FGF10) to derive VBP (groups 1–4). **b** Comparison in expression of SHH, Cytokeratin 8, and stratified cell markers p63, Cytokeratin 5, 14, and 13 between experimental groups and AFE cells that were collected at day 8 of differentiation and received no treatment. Error bars represent ± standard error of the mean, $n = 3$ per group. We used one-way ANOVA to assess statistical significance for tested genes. All of the $p$-values for tested genes are below the threshold $p$-value $< 0.05$ indicating that expression levels of tested genes in FGFs treated groups significantly differed from the expression levels in AFE controls. ACTA Activin A, AFE anterior foregut endoderm, DE definitive endoderm, hiPSC human induced pluripotent stem cells, NG noggin, SB SB-431542, VBP vocal fold basal progenitors. Source data provided as a Source Data file

RNA-seq comparing hiPSC-derived VBP and human adult primary VFEC. EBSeq analysis[37] was applied to identify equally (EEx) and differentially expressed genes (DEx). From 17,523 sequenced genes, 2897 EEx genes with posterior probability (PP) PPEEx ≥ 0.95 (Supplementary Fig. 1), and 6876 DEx genes with PPDEx ≥ 0.95 were found. Enrichment GO biological processes analysis was conducted on each gene set[38,39]. Enriched terms were sorted using score-based ranking to generate top ten enriched GO terms. Our data show that EEx genes were related to formation of apical epithelial surfaces, such as cornification and establishment of the G-protein-coupled receptor signaling pathway that regulates ion channels and transporter proteins (Supplementary Fig. 2a). The top ten DEx enriched terms were involved in the extracellular matrix, fibronectin, and collagen fibril organization (Supplementary Fig. 2b). Differences in gene expression between hiPSC-derived VBP and primary

VFEC may be because VBP have not been exposed to lamina propria and are not yet mature such that hiPSC-derived VBP are in a less differentiated state in 2D culture as compared with mature human primary VFEC.

We compared VBP with human fetal esophageal epithelia. Similar to VF, the esophagus is lined with stratified squamous epithelium that is derived from AFE[24]. We took advantage of a publicly available RNA-seq data set[24] and extracted 18842 genes to calculate fold change value between human fetal esophageal cells and hiPC-derived VBP, to define DEx genes. We found 8843 genes with a fold change >2 or fold change <½ as compared with 6876 DEx genes in hiPSC-derived VBP vs human primary VFEC, suggesting that VBP share a greater degree of similarity with human VF epithelium than human fetal esophageal epithelium. We selected ten genes upregulated in VBP and human adult VFEC, and ten enriched genes upregulated in the fetal esophagus to identify potential tissue-specific markers (Supplementary Table 2). Based on these findings, SOX2 and FOXE1 genes were selected for validation in hiPSC-derived VFEC. FOXE1 gene is upregulated in human fetal esophagus, and was as a marker of esophageal epithelial differentiation[24]. These data suggest that global transcription of VBP is more similar to human VF epithelium than the esophagus and support that VBP are in a less differentiated.

**Generation of a 3D model of hiPSC-derived VF mucosa.** We optimized directed differentiation of hiPSC into functional VF stratified epithelium (Fig. 4a). After derivation of DE (day 4) and AFE (day 8), AFE cultures were treated with high concentration of FGF in DMEM basal medium for 4 days to obtain VBP (day 12). Half-way through VBP differentiation, proliferating VBP were reseeded on collagen–fibroblasts constructs (day 10), attaching to gel and forming a layer. VBP completed their differentiation on the construct in the same medium with high levels of FGF. On day 12, DMEM basal medium was changed for conditional FAD medium supplemented with high levels of FGF and VBP were cultivated as submerged cultures for additional

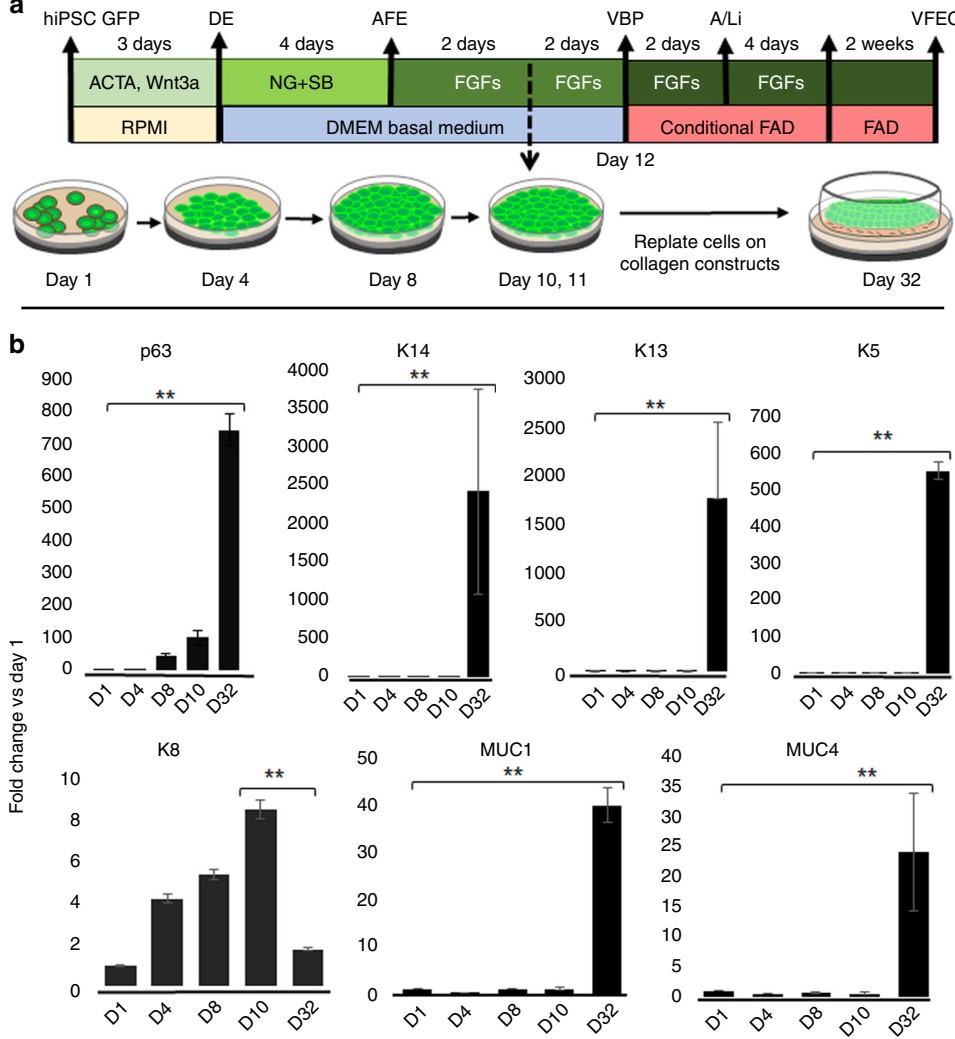

**Fig. 4** Generation of 3D hiPSC-derived human VF mucosa. **a** Protocol for generation of human induced pluripotent stem cell-derived vocal fold mucosa through DE and AFE. **b** Comparison in expression levels of p63, Cytokeratin 14, 13, 5, and 8, and Mucin 1 and 4 between fully differentiated hiPSC-derived VFEC after 32 days in culture and earlier stages of differentiation. VFEC were isolated from the entire constructs at day 32. We used one-way ANOVA to assess statistical significance for tested genes. All of the p-values for tested genes were below the threshold p-value < 0.01 (expressed as **), indicating that expression levels of tested genes in fully differentiated hiPSC-derived VFEC significantly differed from the expression levels of these genes in hiPS cells, DE, AFE, and VBP. ACTA Activin A, AFE anterior foregut endoderm, DE definitive endoderm, hiPSCs human induced pluripotent stem cells, NG Noggin, VBP vocal fold basal progenitors, VFEC vocal fold epithelial cells. Source data provided as a Source Data file

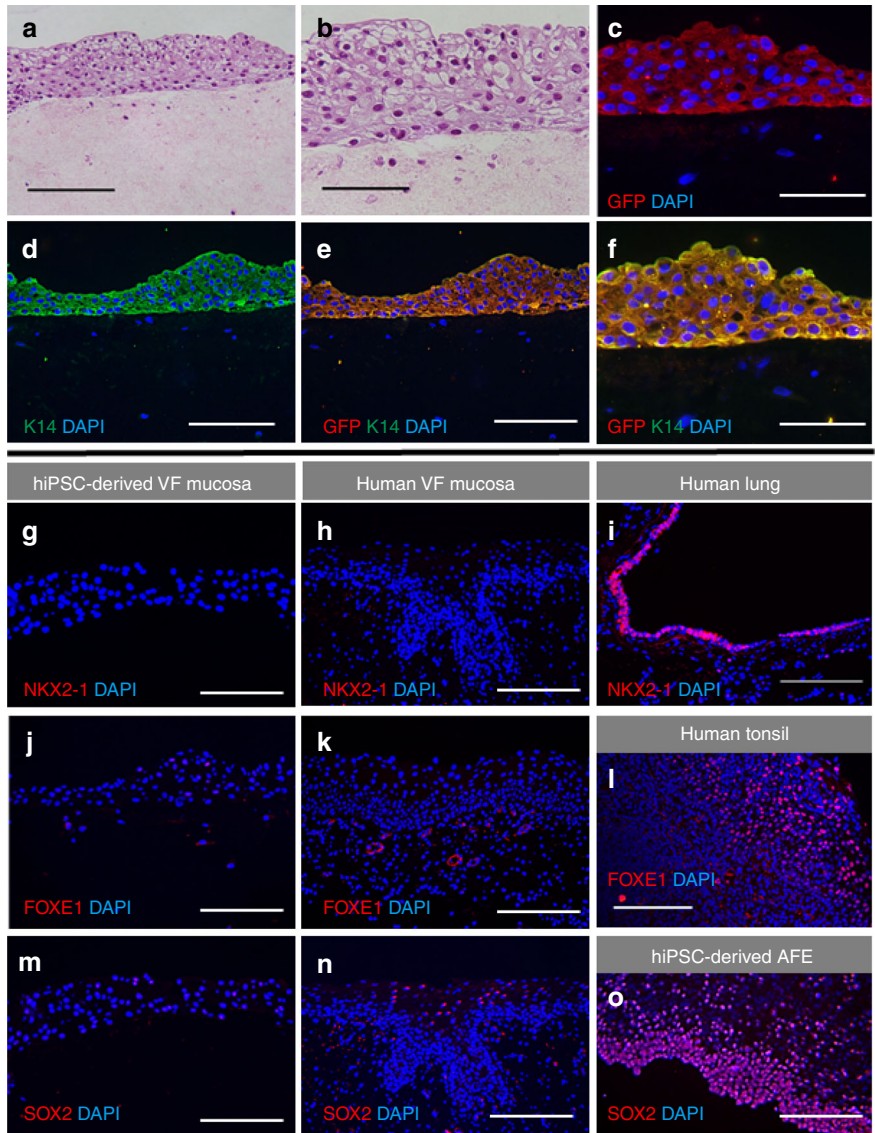

**Fig. 5** Characterization of hiPSC-derived 3D VF mucosa. **a**, **b** Hematoxylin–eosin staining showing morphology of human iPSc-derived VFEC. **c** Immunofluorescent anti-GFP staining of VFEC showing persistency in GFP expression (red). **d** Immunofluorescence staining for Cytokeratin 14 (green). **e**, **f** Double staining of anti-GFP (red) and Cytokeratin 14 (green) showing that GFP + cells preferentially differentiate into Cytokeratin 14 + cells. Scale bars in the panels of **a**, **d**, and **e** = 200 μm. Scale bars in the panels of **b**, **c**, and **f** = 100 μm. **g–o** Confirmation of the specificity of VF epithelial differentiation examining NKX2-1 in red (**g**, **h**), FOXE1 in red (**j**, **k**), and SOX2 in red (**m**, **n**). HiPSC-derived VF epithelium and human VF mucosa were negative for NKX2-1, FOXE1, and SOX2. Human lungs were used as a positive control for NKX2-1 (**i**), human tonsils were used as a positive control for FOXE1 (**l**), and AFE cell cultures were used for confirmation of SOX2 expression (**o**). Scale bars in the panels of **g–o** = 200 μm. AFE anterior foregut endoderm, hiPSC human induced pluripotent stem cell, VF vocal fold

2 days. Conditional FAD with FGF was aspired from the constructs, and an A/Li was created. At the A/Li, cells were cultivated in conditional FAD medium with FGF for 4 days and in plain FAD medium for an additional 2 weeks. After 32 days in culture, engineered VF mucosae were characterized and compared with native human VF mucosae to assess morphology and expression of key VF genes.

We addressed whether replating hiPSC-derived VBP on collagen matrix further promoted stratification, and whether cells were capable to mature in 3D conditions. We isolated cells from whole constructs and performed qPCR analysis to compare expression levels of key stratified epithelial markers p63, K14, K5, and K13 (Fig. 4b). These markers were robustly upregulated in VFEC after 32 days in culture as compared with earlier differentiation stages DE, AFE, or VBP. In contrast, transcript

levels of K8 were significantly downregulated during differentiation on the collagen gel. We compared expression of mucins, MUC1 and MUC4 (Fig. 4b), both of which were significantly upregulated in VFEC after 32 days in culture, suggesting that hiPS-derived epithelium became functional enough to be able to serve as a protective barrier.

Histological characterization of the engineered VF mucosa revealed that after 32 days in culture, hiPSC-derived VFEC created a stratified squamous epithelium, while the collagen gel with human VFF represented lamina propria (Fig. 5a). Epithelium stratified up to ten distinct cell layers (Fig. 5b). HiPSC-derived VFEC formed the epithelium, as confirmed by double staining for anti-GFP and anti-K14 (Fig. 5c–f). Negative staining for NKX2-1 (Fig. 5g, h), FOXE1 (Fig. 5j, k), and SOX2 (Fig. 5m, n) in hiPSC-derived VFEC and native human VF mucosa

demonstrated that hiPCS-derived VF epithelium resembled VF, and not trachea or esophagus.

**Engineered VF mucosa resembles human VF mucosa**. We evaluated morphology, expression of stratified markers and functionality of the hiPSC-derived VF mucosa, and contrasted these to human native VF mucosa. Detailed examination of the morphology of engineered VF mucosa revealed that hiPSC-derived VFEC were mostly cuboidal in shape, and the top cell layer underwent squamation (Fig. 6a). In native VF mucosa, cuboidal cells were also situated in the basal cellular compartment; however, the top cell layer of fully differentiated squamous cells was thicker (Fig. 6b). This difference in morphology can be explained

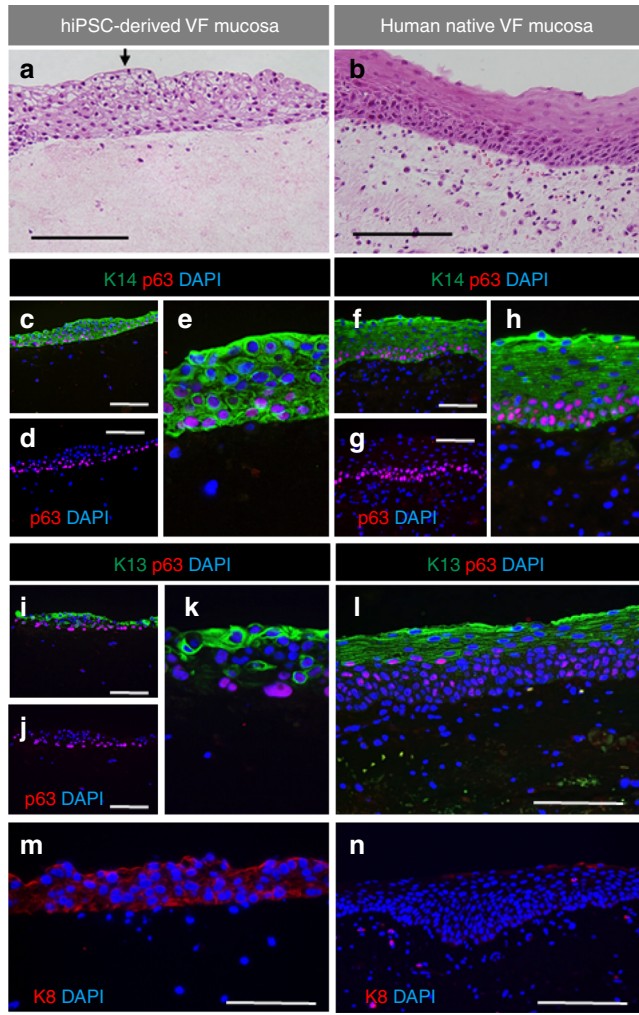

**Fig. 6** Pattern of expression of stratified markers in hiPSC-derived VFEC. **a, b** Hematoxylin–eosin staining showing differences in morphology between hiPSC-derived VFEC (**a**) and human native VF mucosa (**b**). A solid black arrow denotes a flattened luminal cell layer. **c–h** Double immunofluorescent staining showing expression of Cytokeratin 14 (green) and p63 gene (red) in hiPSC-derived VFEC (**c, e**) and native VF mucosa (**f, h**) and anti-p63 staining showing p63 + basal cells in hiPSC-derived VFEC (**d**) and native VF mucosa (**g**). **i–l** Double immunofluorescent staining showing the expression of Cytokeratin 13 (green) along with p63 (red) in hiPSC-derived VFEC (**i, k**) and native VF mucosa (**l**). **j** Anti-p63 staining showing p63 + basal cells in hiPSC-derived VFEC. **m, n** Immunofluorescent staining showing pattern of expression of Cytokeratin 8 in hiPSC-derived VFEC (**m**) and human native VF mucosa (**n**). Scale bar = 200 μm. hiPSC human induced pluripotent stem cell, VF vocal fold

by the fact that native VF mucosa was exposed to mechanical forces secondary to vibration, which could stimulate terminal epithelial cell differentiation.

We further focused on the evaluation of stratification. In both mucosae, p63 gene was found in the basal cell layers along with K14, which diffused into suprabasal cell layers (Fig. 6c–h), while expression of K13 was detected in the suprabasal cell layers only (Fig. 6i–l). This pattern of stratification, observed both in vitro and in vivo conditions, is crucial for proper formation and maintenance of the protective barrier. We detected expression of K8 in hiPSC-derived VFEC, and not in native VF mucosa (Fig. 6m, n). This finding indicates that native VF mucosa obtained from the adult donor was mature.

To assess the compactness of the basement membrane, we performed anti-Laminin alfa 5 (LAMA5) staining. Similar to human VF mucosa, hiPSC-derived VFEC expressed LAMA5 to form the basement membrane anchoring the basal cell layer to the collagen gel (Fig. 7a–c). Expression of the adherence marker E-Cadherin (Ecad) was found throughout the epithelium, while in native VF mucosa it was detected in the suprabasal cellular compartments (Fig. 7d–f).

We examined the functionality of the engineered VF mucosae by measuring mucin expression and capability of VFE to self-renew. In the engineering mucosae and native tissue, Mucin (MUC) 1 was expressed in apical cell layers and formed a superficial mucus coat (Fig. 7g–i). In the engineered VF mucosa, MUC4 was preferentially expressed in the basal cell layer (Fig. 7j, k), while in the native VF mucosa, MUC4 was expressed more superficially (Fig. 7l). For cell proliferation, we found that similar to native VF mucosa, hiPSC-derived VFEC were capable of division; cell proliferation was more intense in the thinner epithelium than in the thicker one with more cell layers (Fig. 7m–o). Overall, our findings demonstrate that engineered VF mucosa resembles native VF mucosa in its structure and function.

**Cigarette smoke extract induces VF mucosa inflammation**. We investigated whether hiPSC-derived VFEC along with primary VFF in collagen matrix were capable of responding to CSE and undergo any structural and/or functional responses. We flooded inserts containing engineered VF mucosae with culture medium supplemented with 5% CSE for 1 week to mimic cigarette smoke exposure[40]. Engineered VF mucosae flooded with plain culture medium were used as negative controls. After 1 week, engineered VF mucosae were collected and analyzed with IF staining and qPCR. qPCR analysis was performed on cells isolated from whole constructs as well as on VFEC and VFF separately, depending upon the genes of interest, to evaluate the effect of cigarette smoke on both cell populations.

We investigated alterations in epithelial morphology. In control VF mucosa, epithelial cells maintained their cuboidal shape with a flattened terminally differentiated top cell layer (Fig. 8a), while in the 5% CSE treated VF mucosa, VFEC were more flattened; epithelium appeared thinner (Fig. 8b). We examined morphology of the basal and luminal cellular compartments; we assayed expression of K14, LAMA5, and K8 (basal layers) and K13 expression (luminal compartment). In control VF mucosa, basal cells expressed p63 and K14 which diffused into superficial cell layers (Fig. 8c–e). In contrast, in the 5% CSE treated VF mucosa, some basal cells downregulated K14, which, accumulated at the luminal cell layers. We observed that some luminal cells appeared to lose their nuclei and fill with K14 only (Fig. 8f–h). To confirm this finding, we performed double staining for GFP and K14. In the control VF mucosa, K14 expressing cells were GFP + (Fig. 8i, j). In the 5% CSE treated VF mucosa, we found some K14-cells located at the basal cellular

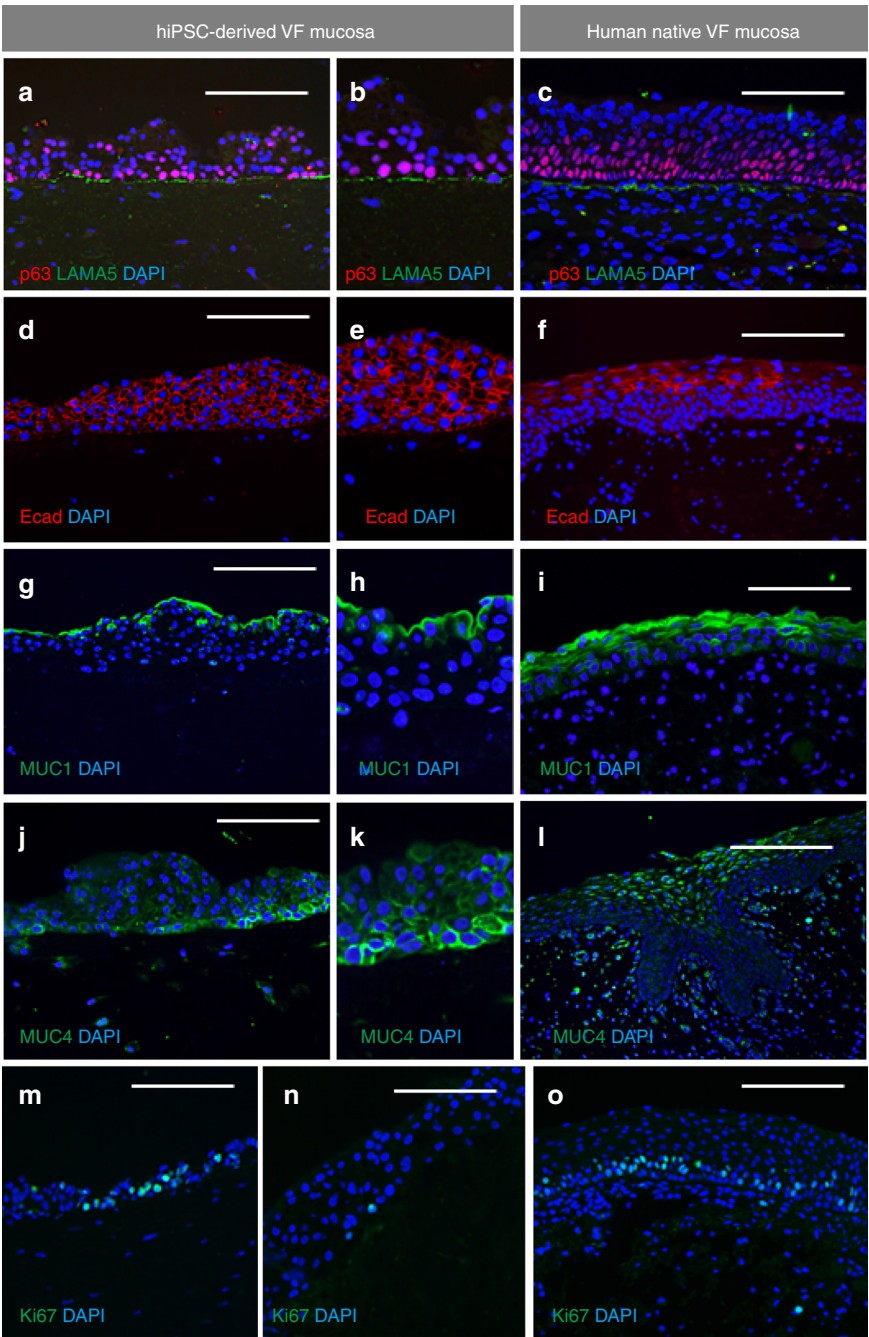

**Fig. 7** Characterization of structural and functional genes expressed in the hiPSC-derived VF mucosa. **a–c** Double immunofluorescent staining for expression p63 (red) and Laminin alfa 5 (green) showing that hiPSC-derived VFEC form a basement membrane (**a**, **b**) similar to VFEC in human native VF mucosa (**c**). **d–f** Immunofluorescent staining comparing the expression pattern of E-Cadherin (red) in hiPSC-derived VF mucosa (**d**, **e**) and human native VF mucosa (**f**). **g–i** Immunofluorescent staining comparing the expression pattern of MUC1 (green) in hiPSC-derived VF mucosa (**g**, **h**) and human native VF mucosa (**i**). **j–l** Immunofluorescent staining comparing the expression pattern of MUC4 (green) in hiPSC-derived VF mucosa (**j**, **k**) and human native VF mucosa (**l**). **m–o** Immunofluorescent staining comparing intensity of cell proliferation, anti-Ki67staining (green), in hiPSC-derived VF mucosa (**m**, **n**), and human native VF mucosa (**o**). Scale bar = 200 μm. hiPSC human induced pluripotent stem cell, VF vocal fold

compartment with weak GFP signal and strong anti-K14 staining in the suprabasal and luminal layers (Fig. 8k, l). Cells in the suprabasal cell layers expressed K13 both in control and 5% CSE treated mucosae (Fig. 8m, n). To assess whether the down-regulation of K14 in the basal cells can trigger conversion of cells into a different cell type, we stained for K8 expression. We found slightly more intense K8 staining in basal cells in the CSE exposed VF mucosa as compared with controls (Fig. 8o–q).

In addition to examining stratification markers, we assayed for LAMA5 and cell Ecad. We found no differences in LAMA5 expression between control and 5% CSE treated cells (Fig. 9a, b). Slightly stronger anti-Ecad staining was detected in the basal cell layer in control VF mucosa as compared with the 5% CSE treated cultures (Fig. 9c, d).

To assess changes in VFEC function, we measured whether exposure to 5% CSE for 1-week stimulated mucin expression,

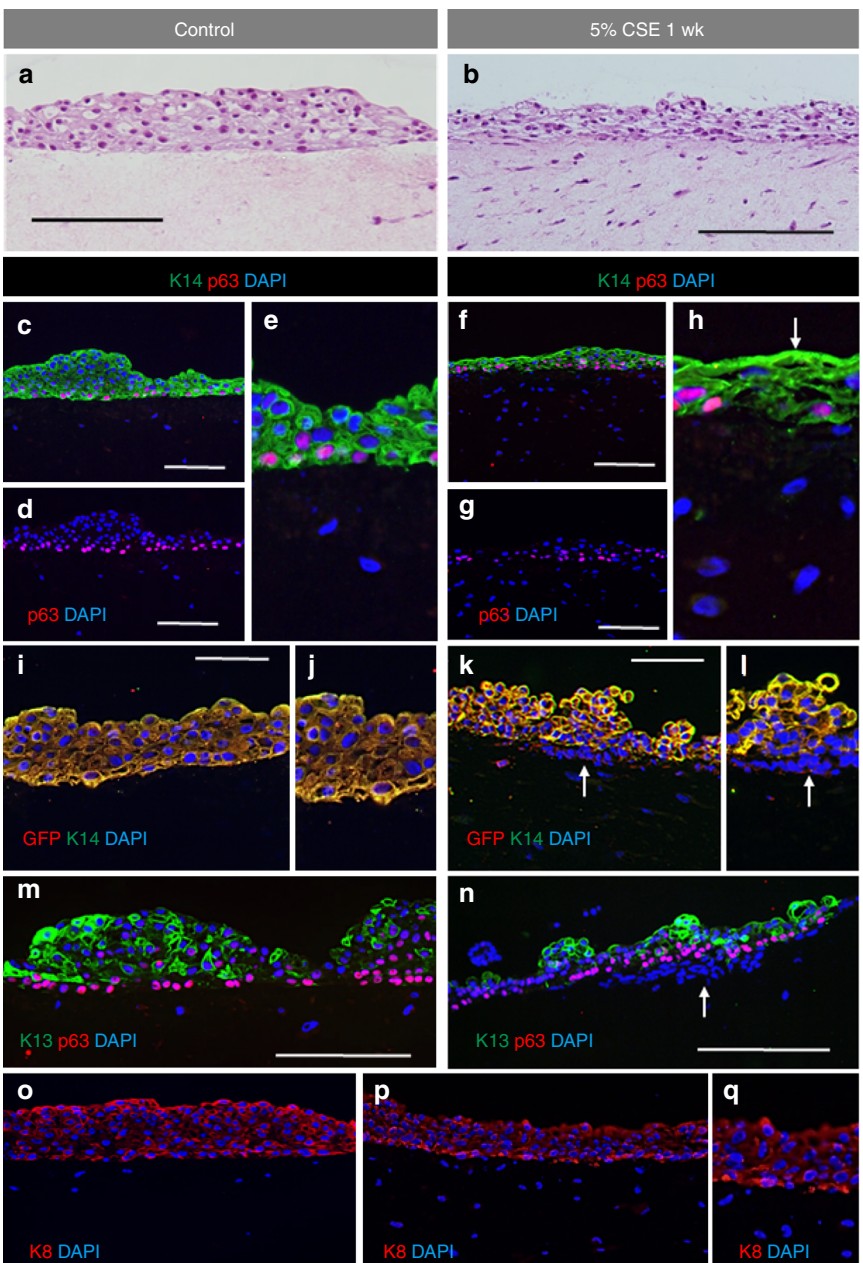

**Fig. 8** Changes in expression of stratified markers after 5% CSE exposure for 1 week. **a**, **b** Hematoxylin–eosin staining showing morphology of the control VF epithelium (**a**) and 5% cigarette smoke extract exposed VF epithelium (**b**). **c–h** Double immunofluorescent staining showing the expression of Cytokeratin 14 (green) and p63 gene (red) in control VF mucosa (**c**, **e**) and CSE exposed VF mucosa (**f**, **h**) and anti-p63 staining showing the p63 + basal cells in control VF mucosa (**d**) and CSE exposed VF mucosa (**g**). **i–l** Double immunofluorescent staining showing the expression of Cytokeratin 14 (green) along with GFP (red) in control VF mucosa (**i**, **j**) and CSE treated VF mucosa (**k**, **l**). White solid arrows point to downregulation of Cytokeratin 14 in the basal cellular compartment. **m**, **n** Double immunofluorescent staining showing the expression of Cytokeratin 13 (green) and p63 gene (red) in control VF mucosa (**m**) and CSE exposed VF mucosa (**n**). **o–q** Immunofluorescent staining showing the pattern of expression of Cytokeratin 8 in control VF mucosa (**o**) and CSE exposed VF mucosa (**p**, **q**). Scale bars in the panels of **a**, **b**, **c**, **d**, **f**, **g**, **m**, **n**, **o**, and **p** = 200 µm. Scale bars in the panels of **i**, **k** = 100 µm. CSE cigarette smoke extract, wk week

MUC1 and MUC4 (Fig. 9e–h) and reduced cell proliferation in VFEC (Fig. 9i, j). Decreases in cell proliferation were confirmed by quantitative measurement of Ki67 + VFEC and VFF in the collagen gel (Fig. 10a). Histological evidence of VF mucosa remodeling were supported by qPCR. Transcript levels of K14, MUC1, and MUC4 were significantly upregulated in CSE treated VF mucosae (Fig. 10c); levels of K13, p63, K8, and Ecad did not change during CSE exposure. We did not observe significant changes in expression levels of structural and functional genes in

VFF such as Collagen 1A1 and 1A2 (Col1A1, Col 1A2), hyaluronan synthase 3 (HAS3), or transforming growth factor beta (TGFbeta), suggesting that VFF were not the primary target for CSE and were protected by VFEC and surrounding extracellular matrix (Fig. 10d). We did measure elevated levels of pro-inflammatory genes, such as the matrix metalloproteinase (MMP-2) (Fig. 10d).

To examine whether both cell types, VFEC and VFF, expressed pro-inflammatory genes, Il-6 and Il-8, we sorted these two-cell

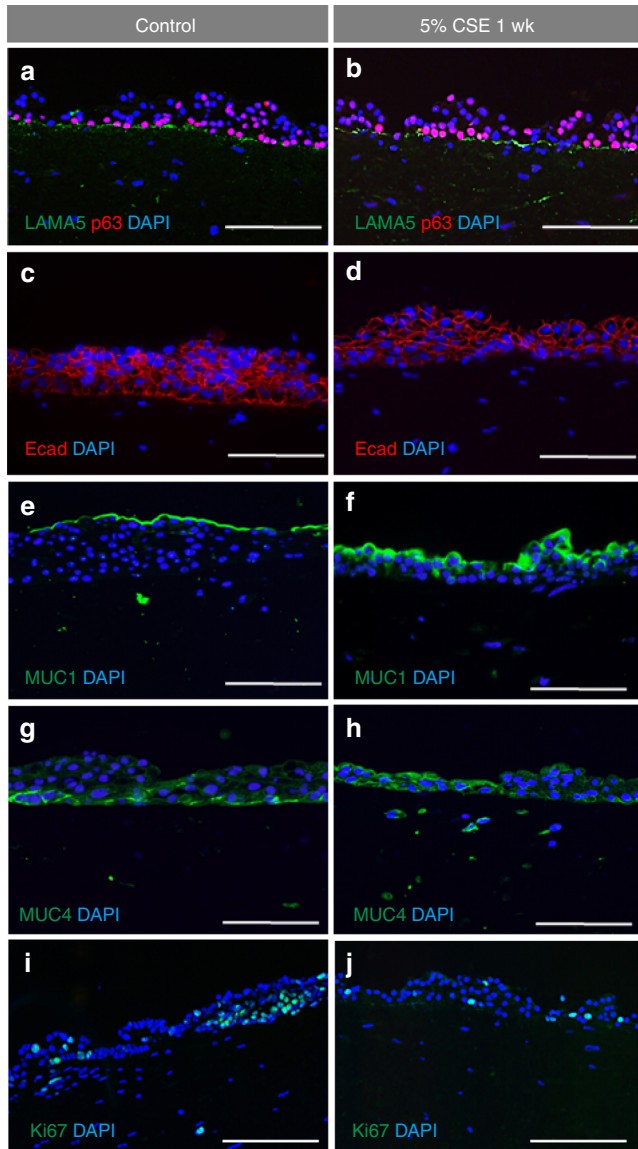

**Fig. 9** Changes in expression structural and functional genes after 5% CSE exposure for 1 week. **a**, **b** Double immunofluorescent staining for expression p63 (red) and Laminin alfa 5 (green) in control VF mucosa (**a**) and CSE VF mucosa (**b**). **c**, **d** Immunofluorescent staining comparing the expression of E- Cadherin (red) in control VF mucosa with slightly more intense staining in the basal cell layer (**c**) and CSE treated VF mucosa (**d**). **e**, **f** Immunofluorescent staining comparing the expression of MUC1 (green) in control VF mucosa (**e**) with more intense MUC1 staining in the CSE VF mucosa (**f**). **g**, **h** Immunofluorescent staining comparing the expression of MUC4 (green) in control VF mucosa (**g**) with a slightly more intense MUC4 staining in the CSE VF mucosa (**h**). **i**, **j** Distribution of Ki67 + VFEC and VFF in control VF mucosa (**i**) and CSE exposed VF mucosa (**j**). CSE cigarette smoke extract, wk week

populations based on their GFP expression using flow cytometry, as both cells types could express these genes. Fluorescent activated cell sorting confirmed that after 1 week of 5% CSE exposure, both cell populations were viable; the number of cells obtained from cell sorting did not significantly differ between control and CSE exposed groups (Fig. 10b). Five percent CSE concentration was, therefore, tolerated by both cell types (Supplementary Fig. 3), as per previous investigations[40].

Separated VFEC and VFF cell populations were analyzed by qPCR (Fig. 10e). We found a robust activation of pro-

inflammatory genes, notably IL-6 in 5% CSE exposed VFEC and IL-8 in 5% CSE exposed VFF, indicating that exposure to 5% CSE-induced mucosal inflammation leading to abnormal VF mucosa remodeling affecting structure and function of the protective epithelial barrier.

## Discussion

In this investigation we developed an in vitro 3D model of human VF mucosa that provides a valuable model for investigation of VF epithelium and fibroblasts in health and disease. This unique system is composed of hiPSC-derived VFEC transfected with GFP and collagen matrix seeded with primary human GFP-negative VFF. Sorting for the presence of GFP enables to dissect the contributions to pathological conditions in both cell types simultaneously. Functional validation of these constructs using 5% CSE revealed that both cell types respond to CSE and undergo morphological and/or functional changes that lead to abnormal VF mucosa remodeling.

Expression patterns of key structural and functional genes in hiPSC-derived VF mucosae was confirmed using human VF mucosal tissue. HiPSC-derived VF mucosae exhibited a slight variability in thickness, varying anywhere from 2 layers up to 10, and had variability in cell proliferation, as cell proliferation is likely correlated to thickness. As for other potential sources of variation, staining levels and distribution of cytokeratins (K13 and K14), LAMA5 and mucins were consistent through different rounds of differentiation. Attempts have been made to generate human sourced in vitro VF mucosae in the past from primary VFEC, however, these constructs fail to re-established stratification in 3D conditions and have lacked a complete basement membrane. These limitations prevent the application of these constructs for translational and pharmacological purposes[20,41] and prevented us from using these specifically to validate VFEC differentiation. To our knowledge, gradual guided differentiation of hiPSC into VFEC that can recapitulate VF epithelial differentiation in vitro represents the only source of cells for organotypic raft cultures that resemble in vivo conditions.

To recapitulate differentiation of stratified squamous VFEC, we built upon our previous findings in mouse models that demonstrated VF basal epithelial progenitors originating from DE and AFE[22]. Recent studies relevant to derivation of keratinocyte or keratinocyte-like epithelial progenitors in vitro have followed epidermal cell lineage differentiation of human embryonic stem (hES) cells using BMP4 and/or retinoic acid (RA) signaling[14,35,42,43]. Since we used hiPSC instead of hES cells and wanted recapitulation of the development of VFEC, selection of RA and BMP4 for differentiation of VBP did not lead to efficient stratification. In foregut-derived progenitors, selection of RA along with BMP4 can posteriorize and at the same time dorsalize cell fate toward a PDX1-positive pancreatic duodenal cell type[44], which is in accordance with our observations. To induce stratification in AFE-derived VBP, we focused on modulation of the FGF signaling pathway, which is a downstream target of Wnt and BMP4[45]. Our previous studies demonstrate that Wnt signaling is active during the establishment of VF basal progenitors in vivo and conditional deletion of β-catenin in VF basal epithelial progenitors affects epithelial stratification[23]. We utilized this knowledge and increased concentration of FGF in culture media, which promoted expression of stratified makers in AFE-derived VBP. Our data show that increases in expression of stratified markers in AFE cells were independent on SHH signaling. This correlates with our in vivo studies[22] that demonstrate that the role of SHH signaling is necessary for early specification of laryngopharyngeal endoderm rather than establishment of basal progenitors and their stratification[22]. The final conversion of

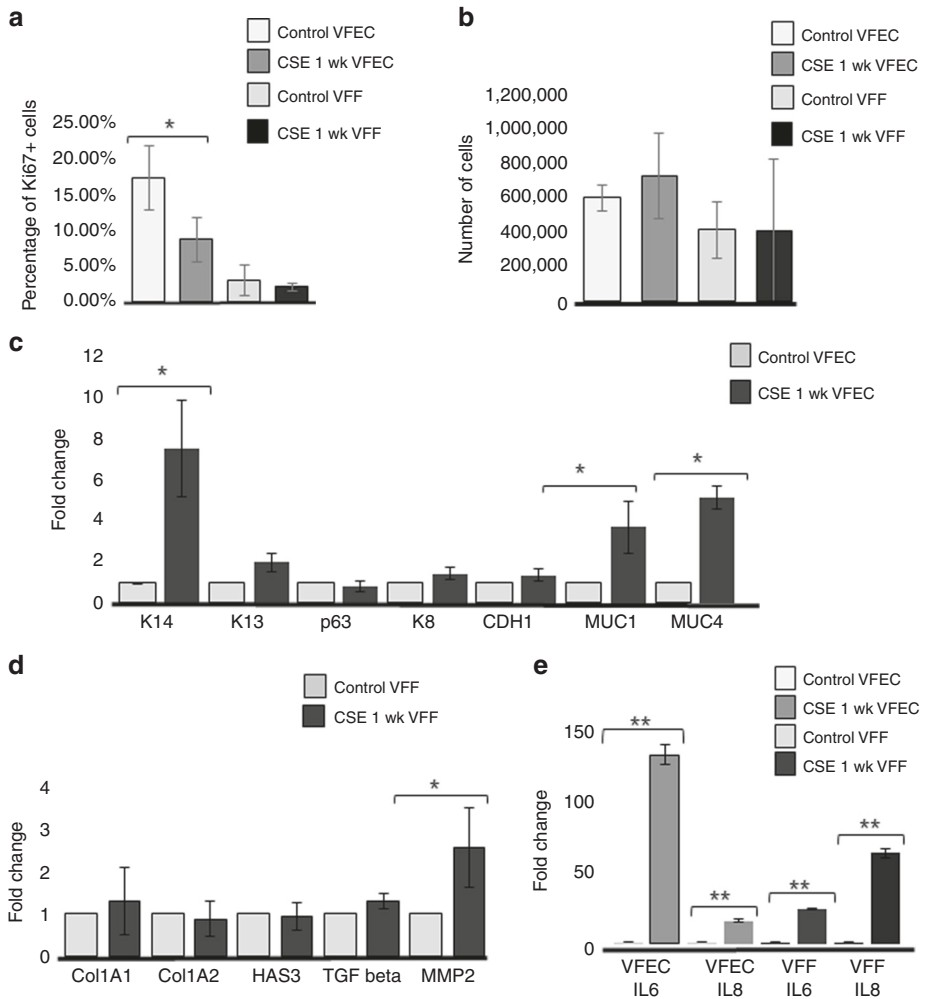

**Fig. 10** Transcript levels of characteristic stratified and epithelial markers after 5% CSE exposure for 1 week. **a** Quantitative assessment of cell proliferation in control VFEC and VFF vs VFEC and VFF exposed to the CSE. Error bars represent ± standard error of the mean, $n = 3$. Two-sample $t$ tests confirmed statistical significant decreases in cell proliferation in CSE exposed VFEC only ($p < 0.05$, expressed as *). **b** Quantitative assessment of the cell number sorted by flow cytometry. Error bars represent ± standard error of the mean, $n = 3$. Two-sample $t$ tests showed that there was no significant difference in cell numbers between control and CSE exposed VFEC and VFF ($p$-values were above the threshold, $p > 0.05$), indicating that CSE does not affect the viability. **c** Changes in the expression levels of selected epithelial genes: Cytokeratin 14, 13, 8, p63 gene, CDH1 (E-Cadherin), MUC1, MUC4 in control VFEC, and 5% CSE exposed VFEC. Cells were isolated from entire constructs in control and 5% CSE exposed experimental groups. We used two-sample $t$ tests to assess statistical significance in tested genes. Increased expression levels of Cytokeratin 14, MUC1, and MUC4 in 5% CSE treated group were statistically significant ($p < 0.05$, expressed as *). Error bars represent ± standard error of the mean, $n = 3$. **d** Changes in expression levels in structural (Col1A1; Col1A2; HAS3) and functional (TGF beta and MMP-2) genes in VFF. Cells were isolated from entire constructs in control and 5% CSE exposed experimental groups. Two-sample $t$ tests confirmed statistical significance in MMP-2 expression only ($p < 0.05$, expressed as *). $P$-values of other genes were above $p > 0.05$, suggesting the 5% CSE did not significantly change the morphology or function of VFF. Error bars represent ± standard error of the mean, $n = 3$. **e** Activation of pro-inflammatory genes, Il6 and Il8, in control and CSE exposed VFEC and VFF. VFEC and VFF were analyzed and evaluated independently, as they were sorted by the flow cytometry prior to RNA isolation. Two-sample $t$ tests confirmed statistical significance in expression in pro-inflammatory genes in cigarette smoke extract exposed VFEC and VFF ($p < 0.01$, expressed as **). CSE cigarette smoke extract, VFEC vocal fold epithelial cells, VFF vocal fold fibroblasts, wk. week. Source data provided as a Source Data file

hiPSC-derived VBP into functional VFEC was performed by replating VBP on top of collagen constructs seeded with primary VFF and their cultivation at an A/LI. The primary VFF cell line was developed from normal vocal folds obtained from a 21-year-old male donor[46]. This cell line has been shown to support proper differentiation of keratinocyte-like cells and their stratification at an A/LI[14,42]. Use of conditional FAD medium along with co-culture of hiPSC-derived VFEC with a sufficient number of primary VFF as feeders in collagen matrix was important to eliminate possible heterogeneity among cells during iPS cell differentiation, as demonstrated with RNA sequencing and retain epithelial stratification at the A/LI specific. These findings are in

accordance with the literature regarding generation of skin organotypic raft cultures from human primary and ESC-derived keratinocytes whereby confirming that selection and number of fibroblasts in the feeding collagen gel significantly influence the quality of epithelial differentiation[19,34,47,48].

To test functionality of the VF organotypic mucosa model, we exposed engineered VF mucosae to 5% CSE for 1 week to determine if we can induce mucosal inflammation and simulate abnormal keratotic changes in the VF mucosa that are related to smoking. The primary goal was to reveal changes of evaluated markers induced by CSE that can be potentially utilized to modulate responses of VFEC and VFF to cigarette smoke,

in vitro. Clincally, histological characteristics of keratotic lesions have been described[49–54]. Keratosis means total replacement of superficial epithelial cells by cytokeratin filaments and dissolution of the nuclei, irregular stratification and loss of epithelial polarity ultimately leading to increased basal cell proliferation[49,52,53,55]. Molecular data obtained from biopsies in patients with dysplastic lesions confirmed 8.53-fold increase in expression of insulin-like growth factor 1 and significant increase in pro-inflammatory genes, such as matrix metalloproteinase-2 (MMP-2), or S100 calcium-binding protein A4[56]. We show that the hiPSC-derived 3D models of human VF mucosa could provide significant information regarding genetic and molecular regulation of abnormal VF mucosa remodeling in response to mucosal inflammation, including identification of targets for possible genetic or pharmacological manipulations. These targets include compromised apico–basal epithelial polarity as confirmed with downregulation of K14 and upregulation of K8 in basal layers and superficial pathological accumulation of K14, impaired mucus production and activation of pro-inflammatory genes in epithelial cells and fibroblasts (IL6, IL8, and MMP-2). Such studies may accelerate the discovery of therapeutics to modulate the responses of VFEC and /or VFF and improving their repair or focusing on anti-inflammatory products.

In summary, this investigation provides a framework for the efficient generation physiologically relevant and clinically useful developmentally derived VF mucosa that can be used as a 3D in vitro system for VF mucosal disease modeling and testing therapeutic approaches for the treatment of vocal fold remodeling, inflammation or other laryngeal diseases.

## Methods

**Statement on ethics**. All stem cell work in this investigation was approved by the Stem Cell Research Oversight Committee at the University of Wisconsin Madison (SC-2015-0008).

The University of Wisconsin Madison Institutional Review Board approved the collection of human larynges under Protocol (2017-0885).

**Targeting AAVS1 locus of hiPSC with Talens and eGFP reporter**. All stem cell work in this investigation was approved by the Stem Cell Research Oversight Committee at the University of Wisconsin Madison (SC-2015-0008). The enhanced GFP reporter gene was targeted to the AAVS1 locus by homologous recombination in IMR-90-4 hiPSC using AAVS1 TALENs and an eGFP donor plasmid with puromycin selection cassette[21]. Briefly, human iPS IMR-90-4 cells were cultured and expanded in Matrigel (BD Biosciences, San Jose, CA) coated six-well plates in mTesr1 media (WiCell, Madison, WI). When reaching 60% confluency, hiPS cells were digested with pre-warmed Accutase (StemCell Technologies, Vancouver, CA) (1 ml/well) for 5 min room temperature (RT), then detached and collected in five times volume of phosphate buffered solution (PBS). Cells were centrifuged with 10,000 rpm for 3 min and resuspended in 100 μl of P3 nucleofection solution (P3 Primary Cell 4D-Nucleofector X kit; Lonza, Basel, Switzerland) with 10 μg of AAV-CAGGS-EGFP donor (Addgene, Cambridge, MA, USA; plasmid # 22212) and 5 μg of each AAVS1-TALEN plasmids (Addgene, Cambridge, MA, USA; L1 plasmid #: 35431; R1 plasmid #: 35432). The total volume of plasmids did not exceed 10% of transfection solution as recommended by Luo et al.[21]. Cells were transfected in 4D-Nucleofector TM Core Unit System using program CM-120. Transfected cells were then plated on Matrigel and cultured in NutriStem medium (Stemgnet, Lexington, MA, USA) supplemented with Y-27632 (10 μM) (Tocris Bioscience, Minneapolis, MN, USA) to slow down the growth of transfected cell colonies and to stimulate their survival during drug selection. Transfection efficiency was 70% (Fig. 1a). Drug selection was initiated next day in NutriStem medium containing 0.3 μg/ml puromycin for 2 days, with changing medium daily. We continued drug selection in NutriStem medium with puromycin 0.5 μg/ml for another 8 days until colonies grew big enough to be picked up. After day 10, stably transfected colonies were picked up, digested with Accutase and transferred into 96-well plate with Matrigel in mTesr1 media supplemented with Y-27632 (10 μM) for clone formation. MTesr1 medium was changed every day until positive clones reached confluence ~80%. Stably transfected positive clones were then passaged into 24-well plates coated with Matrigel for further expansion, PCR screening and southern blot analysis (Fig. 1b; Supplementary Fig. 4). Both PCR screening and southern blot were performed to confirm AAVS1-targeted GFP integration as per protocol;[21] sequences of primers for PCR and southern blot are included in Supplementary Table 1. Based on our results, a clone B10 was selected for further differentiation (Fig. 1c–k).

**Human iPS-GFP cell culture and differentiation**. Human iPS-GFP IMR-90-4 reporter cells were maintained in an undifferentiated state in mTesr1 media on plates coated with Matrigel, and they were routinely passaged with Versene (StemCell Technologies, Vancouver, CA) in a ratio of 1:6. When cells reached 80% confluency, DE induction was performed using RPMI medium with Glutamax (Gibco, Life Technologies) supplemented with 100 ng/ml Activin A (Peprotech, Rocky Hill, NJ, USA) and 25 ng/ml Wnt 3a (R&D System, Minneapolis, MN, USA) for 1 day and RPMI media with Glutamax supplemented with 100 ng/ml Activin A and 0.2% fetal bovine serum (FBS) for additional 2 days[25,32].

AFE differentiation was performed[25,26] starting with day 4, RPMI medium was replaced by DMEM/F12 medium with Glutamax (Gibco, Life Technologies) supplemented with N2 and B27 supplements (Gibco, Life Technologies), ascorbic acid 0.05 mg/ml (Millipore Sigma, St. Louis, MO), monothioglycerol (MTG) 0.4 mM (Millipore Sigma, St. Louis, MO) (here refer as DMEM basal medium), 200 ng/ml Noggin (R&D System, Minneapolis, MN, USA), and 10 μM SB431542 (Tocris, Minneapolis, MN, USA) for 4 days to induce the AFE formation. Medium was changed daily.

At day 8, AFE-derived cells were differentiated into the VBP to induce expression of stratified markers for additional 4 days with medium changed every other day. We used DMEM basal medium supplemented with 1% penicillin–streptomycin (Invitrogen, Carlsbad, CA, USA) and a combination of FGF growth factors based on experimental conditions. In group 2, FGF2 (50 ng/ml), group 3 – FGF2 (250 ng/ml), and group 4 (FGF2 250 ng/ml; FGF7 100 ng/ml; FGF10 100 ng/ml). FGFs signals were purchased from R&D System, Minneapolis, MN, USA. At day 12, the VFB were detached with 0.05% TE trypsin in EDTA (Gibco Life Technologies) and used for RNA isolation and qPCR analysis. For cell reseding on collagen consructs, VBP were mildly trrypsinized with 0.025% TE trypsin and reseeded at days 10 -11.

VF mucosa constructs were prepared by combining high concentration rat tail collagen (4 mg/ml; 80% final volume BD Biosciences) and 10xDMEM (10% final volume; Millipore Sigma, St. Louis, MO, USA) on ice and adjusting pH with 1 N NaOH (pH=7.2 - 7.4). VF primary fibroblasts 21T cells, passage P5 - P6, were resuspended in ice-cold FBS (10% final volume; 500,000 cells/ml final volume) and added to a collagen mixture as described by others (14, 35, and 43). A mixture of collagen gel and VF fibroblasts was plated on transwell cell culture inserts (Corning, Millipore Sigma, St. Louis, MO, USA), 2 ml per a six-well culture insert, and solidified for 1 hour (h) in a tissue incubator at 5% $CO_2$, 37 °C. After 1 h, collagen was gently detached with a pauster pipette and constructs were flooded with DMEM basal medium, returned into an incubator and left at least 24 h to allow for gel contraction. The next day, the VBP were mildly trypsinized and plated on collagen constructs at high density in 100 μl DMEM basal medium supplemented with high concentration of FGF2 (250 ng/ml), FGF10 (100 ng/ml), and FGF7 (100 ng/ml). Cells were allowed to attach for at least 2 h. Then they were flooded with DMEM basal medium supplemented with high levels of FGFs as mentioned above, and cultivated for additional 2 days to complete VBP differentiation (day 12). VBP were collected 2–3 days after inducing stratification in DMEM basal medium supplemented with high levels of FGFs as mentioned above, at days 10–11. We determined that for successful reseeding of cells on the construct, VBP need to be fully confluent but still proliferating to attach to the gel and form an even cell layer. Only cell cultures that fulfilled these criteria were suitable for reseeding on collagen gel. On the other hand, during the prolong cultivation of cells in 2D conditions up to day 12, VBP become too confluent and proliferation slowed.

On day 12 or 13 (depending on the timing of VBP reseeding), medium was changed for conditional FAD medium supplemented with high levels of FGFs, and the VBP were further differentiated as submerged cultures for 2 days and then at the A/Li. The A/Li was performed in conditional FAD medium with FGFs for first 4 days and plain FAD medium for additional 2 weeks. FAD medium was freshly prepared every week. It consisted of the DMEM medium and F12 in ratio 1:3 (Gibco Life Technologies), supplemented with 2.5 ml FBS, 0.4 μg/ml hydrocortisone (Millipore Sigma, St. Louis, MO, USA), 8.4 ng /ml cholera toxin (Millipore Sigma, St. Louis, MO, USA), 5 μg/ml insulin (Millipore Sigma, St. Louis, MO, USA), 24 μg/ml adenine (Millipore Sigma, St. Louis, MO, USA), 10 ng/ml epidermal growth factor, 1% penicillin–streptomycin (Invitrogen, Carlsbad, CA, USA). In submerged cultures, 1 ml of FAD was applied on the transwells with collagen constructs, and 2 ml were applied in the basolateral chamber. FAD was changed every other day. To create the A/Li, FAD medium was placed in the basolateral chamber only and changed three times a week. Conditional FAD medium was formed by cultivation of FAD with human primary VFF 21T cells for 24 h in 37 °C in 5% $CO_2$-humidified atmosphere. After 24 h the medium was collected, sterile-filtered, and stored at −20 °C. The ratio of 30:70 (30% for conditional and 70% for fresh FAD medium) was used in the experiment. A step-by-step protocol describing our derivation of hiPSC-derived vocal fold mucosa can be found in Nature Protocol Exchange[57].

**Characterization of hiPSC during differentiation**. Human iPS cells were characterized during differentiation using immunohistochemistry, flow cytometry, and quantitative polymerase chain reaction (qPCR). Cells cultured on six-well plates were fixed for 10 min with fresh 4% paraformaldehyde and permeabilized with PBS containing 0.1% Triton X-100 for 20 min. Cells were then washed with PBS twice

and blocked with blocking solution composed of 8.8 ml PBS, 0.5 ml goat serum, 0.3 ml 1% Triton X-100, and 0.4 ml 25% bovine serum albumin (BSA) for 30 min at room temperature (RT), and incubated with primary antibodies in blocking solution at 4 °C overnight. Staining was detected with Alexa fluorophore-conjugated secondary antibodies, diluted in blocking solution (1:500), and cultivated for 1.5 h at RT. Cells were washed with PBS twice and mounted using Vectashield with DAPI (Vector Laboratories; Peterborough, UK). Images were captured using an EVOS FL inverted fluorescent microscope. Primary antibodies used included anti-rabbit FOXA2 diluted to 1:1000 (Abcam, Cambridge, CA, USA), anti-goat SOX17 antibody diluted to 1:200 (R&D System, Minneapolis, MN, USA), anti-mouse EPCAM diluted to 1:100 (StemCell Technologies, Vancouver, CA), anti-rabbit SOX2 antibody diluted to 1:200 (Abcam, Cambridge, CA, USA), anti-rabbit cytokeratin K8 diluted to 1:250 (LS Bio, Seattle, WA, USA) and anti-rabbit TAFp63 diluted to 1:100 (H-300) (Santa Cruz Biotechnology, Dallas, TX, USA). The secondary antibody used was Alexa Fluor TM 594 goat anti-rabbit Ab, Alexa Fluor 594 donkey anti-goat Ab and Alexa Fluor TM 488 goat anti-mouse Ab (all Invitrogen, Carlsbad, CA, USA). Parallel wells stained with the secondary antibody alone were included as negative controls in each experiment.

Collagen gel constructs were first washed in PBS, fixed in fresh 4% paraformaldehyde for 15 min at RT, and embedded in histogel (ThermoFisher Scientific, Waltham, MA, USA). Constructs were dehydrated in series of ethanol, treated with xylene, embedded in paraffin, and cut to 5 -μm thick serial sections. Sections were then deparaffinized, rehydrated, and stained using a standard IHC protocol[22]. Antigen retrieval was performed by heating sections in sodium citrate pH = 6 at 80 °C water bath for 2 h. Primary antibodies used included anti-goat GFP-Tag diluted to 1:500 (ThermoFisher Scietific, Waltham, MA, USA), anti-rabbit Ki67 diluted to 1:300 (Abcam, Cambridge, CA, USA), anti-rabbit cytokeratin K8 diluted to 1:250 (LS Bio, Seattle, WA, USA), rabbit anti-Laminin alfa 5 diluted at 1:100 (Abcam, Cambridge, UK), anti-rabbit cytokeratin K14 diluted to 1:250 (Proteintech, Rosemont, IL, USA), anti-rabbit cytokeratin K13 diluted to 1:200 (Abcam, Cambridge, CA, USA), anti-rabbit E-Cadherin diluted to 1:250 (Cell Signaling, Danvers, MA, USA), anti-mouse p63 diluted to 1:100 (Santa Cruz Biotechnology, Dallas, TX, USA), and anti-mouse MUC1 and MUC4 diluted to 1:200 (both Abcam, Cambridge, CA, USA), anti-mouse NKX2-1 diluted to 1:100 (Lab Vision Corporation, Fremont, CA, USA), and anti-goat FOXE1 diluted to 1:100 (Abcam, Cambridge, CA, USA). Secondary antibodies used were Alexa Fluor TM 594 goat anti-rabbit (Invitrogen, Carlsbad, CA, USA) at the dilution 1:500, Alexa Fluor 594 donkey anti-goat Ab diluted to 1:500 (Invitrogen, Carlsbad, CA, USA), Alexa Fluor 488 goat anti-mouse Ab (Invitrogen, Carlsbad, CA, USA) at dilution 1:500, FITC-conjugated goat anti-rabbit Ab, at the dilution 1:100 and Cy5-cojugated goat anti-mouse at the dilution 1:200 (both Jackson ImmunoResearch, West Grove, PA, USA). They were applied 1 h and 30 min at RT. Slides were mounted using Vectashield with DAPI (Vector Laboratories, Peterborough, UK). Images were taken with a Nikon Eclipse E600 with a camera Olympus DP 71, and were adjusted for brightness using the installed DP 71 software, Olympus Corporation.

Ki67 + and DAPI (Ki67 −)-labeled VFEC and VFF were manually counted using Image J. For each experiment, ten sections from three different differentiation rounds were analyzed (three control VF mucosae, n = 30 sections and three 5% CSE exposed VF mucosae, n = 30). Percentage of Ki67 + nuclei in VFEC and VFF in control and experimental groups was compared using the Student's t test. The results are reported as mean ± s.d., and were considered statistically significant at p ≤ 0.05.

For flow cytometry, cells were dissociated with Accutase for 5 min at 37 °C, centrifuged, and washed in washing buffer (PBS, 2% FBS). They were then centrifuged and resuspended in 100 μl of washing buffer. One test volume of antibody was added for each 100 μl cell suspension. BV421 mouse anti-human CD184 (also known as CXCR4) and PE anti-EpCAM (EBA-1), both obtained from BD Biosciences (BD Biosciences, San Jose, CA, USA), were added to DE samples. APC-R700 mouse anti-human CD-56 (BD Biosciences, San Jose, CA, USA) and APC anti-CD271 (also known as NGFR) (Miltenyi Biotec, Bergisch Gladbach, GE) were added to AFE cell cultures. Cells were stained for 30 min on ice, washed in 3 ml of washing buffer (PBS and 2% FBS) once, centrifuged, and resuspended in FACS buffer (PBS, 10 mM EDTA, and 2% FBS). Before sorting, cells were passed through a 40-μm cell strainer. Cells were analyzed and sorted with a FACSAria2 (BD Biosciences, San Jose, CA, USA). Compensation beads with one test volume of each antibody were used as negative controls. FACS data were analyzed with J Flow software (BD Biosciences, San Jose, CA, USA). The data are presented as the average of the three biological replicates ± s.e.m.

For qPCR, hiPSC cells at different stages of differentiation (iPS cells, DE, AFE, VBP, and VFEC) were dissociated with 0.05% TE Trypsin-Edta (Gibco, Life Technologies Corporation) and lysed in preparation for RNA isolation. The total RNA was extracted from cells using the RNeasy Mini and Micro Kit (Qiagen, Valencia, CA) following the manufacturer's protocols. One thousand nanograms of RNA was reverse transcribed to cDNA using reverse transcription reagents (Go Script, Promega, Madison, WI, USA) per the manufacturer's protocol. The total volume of 0.4 μl of cDNA was used per 20 μl real-time qPCR reaction using Power Up Sybr Green Master Mix (Applied Biosystem, Foster City, CA, USA) and run for 40 cycles in triplicates on a 7500 Fast Real Time PCR System machine (Applied Biosystem, Foster City, CA, USA), according to the manufacturer's instructions. Gene-specific primers are listed in Supplementary Table 1. Relative gene expression, normalized to beta-actin, and undifferentiated iPS cells, AFE, or human primary cell controls (ΔΔCt), was calculated as fold change using the 2(-ΔΔCt) method[58]. If undetected, a cycle number 40 was assigned to allow fold change calculation. The data are presented as the average of the three biological replicates ± s.e.m.

For primary epithelial culture, normal vocal folds of two human adults (female, 67 and 80 years old) obtained from autopsy were used in this study. The University of Wisconsin Madison Institutional Review Board approved the collection of human larynges under Protocol (2017-0885). In both cases, larynges and vocal folds were normal and did not have any disease. Larynges were received within 28 h postmortem. For primary cell culture, the epithelial layer of true vocal fold was dissected and cut into small pieces and suspended in Airway Epithelial Cell Medium (ATCC, Manassas, VA, USA) with 25 ng/ml cholera toxin (Sigma, St. Louis, MO, USA) and 100 ng/ml primocin (InvivoGen, San Diego, CA, USA). Cells were grown on collagen–fibronectin-coated tissue culture dishes at 37 °C in 5% $CO_2$-humidified atmosphere. After 2–3 weeks, the adherent epithelial cells were trypsinized (0.05% Trypsin-0.02% EDTA, ATCC) and passaged several times to obtain pure epithelial cell population.

For RNA sequencing, VBP treated for 4 days in the mix of FGFs and human primary cells were dissociated with 0.05% TE Trypsin-EDTA (Gibco, Life Technologies Corporation) and lysed in preparation for RNA isolation. The total RNA was extracted from cells using the RNeasy Mini and Micro Kit (Qiagen, Valencia, CA) following the manufacturer's protocols. RNA sequencing was performed by the University of Wisconsin-Madison Biotechnology Gene Expression Center & DNA Sequencing Facility. Quality control was done in FastQC v0.11.5 using standard defaults;[58] all samples passed and were used in downstream analysis. Reads were mapped back to the genome using the short-read aligner Bowtie v1.0.0[59], followed by RSEM v1.2.7[60] to estimate gene expression. R/EBSeq[37] was applied to identify genes as differentially or equivalently expressed (DEx or EEx, respectively). EBSeq is an empirical Bayesian approach that models a number of features observed in RNA-seq data. It compares the latent level of expression in group 1 ($\mu$ 1; normal human cell) with that in group 2 ($\mu$ 2; iPS cell) accounting for gene-specific differences in average expression level and variability; and calculates the posterior probability of a gene being DEx (e.g., $\mu$ 1 ≠ $\mu$ 2) or EEx (e.g., $\mu$ 1 = $\mu$ 2). A gene is labeled DEx if the posterior probability of DEx is >0.95, which controls the overall false discovery rate (FDR) at 5%. Enrichment of common functions was assessed using Enrichr[37,38]. Two biological replicates and two technical replicates (n = 4) were used for RNA sequencing comparisons. Primary human VFEC were isolated from two different donors; hiPSC-derived VBP were obtained from two different differentiation rounds.

**Functional evaluation of hiPSC-derived VF mucosa**. Cigarette smoke extract (CSE) (100%) was generated[40]. Briefly, mainstream smoke of Research-grade cigarettes (3R4F) with filter (Kentucky Tobacco Research and Development Center at the University of Kentucky; Lexington, KY) was bubbled through 35 ml the DMEM/F12 medium (Gibco Technologies) in a disposable 50 -ml tube with the use of an experimenter-operated syringe. Human smoking was modeled with short puffs (2 s) with long delays between puffs (30 s). The mainstream smoke, smoke that is drawn through the end of the cigarette during puffing, was bubbled through the DMEM medium. Three cigarettes were "smoked" until a cigarette filter. The obtained medium was considered as 100% CSE[40]. To ensure standardization between experiments, CSE was sterile-filtered through a 0.2 -mm filter, aliquoted, and stored at −80 °C. For usage, CSE was quickly thawed and diluted with the FAD medium to the indicated concentration and used the same day.

To mimic exposure of VF mucosa to the CSE, inserts (upper chamber) containing engineered VF mucosa were flooded with FAD medium supplemented with 5% CSE for 1 week. The lower chamber was flooded with plain FAD medium. Medium was changed every day in both chambers. Engineered VF mucosa flooded with plain FAD medium were used as negative controls in the experiment. After 1 week of exposure to the 5% CSE, engineered VF mucosae were characterized with IHC and qPCR to investigate expression levels of clinically relevant genes in GFP + VFEC and GFP − VFF.

To isolate populations of hiPSC-derived VFEC and VFF from the collagen–fibroblast constructs, the collagen gel was first dissolved using collagenase (Gibco™ Collagenase, Type I, Powder; 17018029 Gibco™). Briefly, old medium was aspirated, and constructs were washed twice in PBS. Collagenase type I at working concentration 100 U/ml was added to the upper (1 ml) and lower chambers (2 ml), with VF mucosa being fully submerged. Constructs were incubated at 37 °C for at least 2–3 h, or until the collagen completely dissolved and cells got loose. The cell suspension was then transferred into a 15 -ml conical tube. Cells were centrifuged for 5 min at 10,000 rpm, resuspended in PBS, and transferred into a 1.5 -ml tube. Cells were again centrifuged for 5 min at 10,000 rpm, the supernatant was aspirated, and cell pellet was stored at −80 °C.

For separation of GFP + (VFEC) and GFP− (VFF) cell populations by flow cytometry, cells were first isolated using collagenase as described above, and transferred into a 15 -ml conical tube. Cells were centrifuged for 5 min at 10,000 rpm and washed in 3 ml of washing buffer (PBS and 0.1% bovine albumin–BSA), centrifuged for 5 min at 10,000 rpm and resuspended in the sorting FACS buffer (PBS, 10 mM EDTA, and 0.1% BSA). Before cell sorting, the cell suspension was filtered through a 40-μm cell strainer. Cells were sorted based on

the presence of GFP with a FACSAria2 sorter (BD Biosciences, San Jose, CA, USA). Primary GFP− VFF were used as negative controls in this sorting. GFP− and GFP + cell populations were collected into PBS 0.1% BSA. GFP + and GFP− cell populations were then centrifuged, medium was aspired, and cells were stored at −80 °C.

Finally, cells isolated from the whole constructs or separated GFP + VFEC and GFP− VFF obtained from the cell sorting were used for RNA isolation using ReliaPrep™ RNACell Miniprep System (Promega, Madison, WI, USA) following the manufacturer's protocol, Five hundred nanograms of RNA were reverse transcribed to cDNA using reverse transcription reagents (Go Script, Promega, Madison, WI, USA) per the manufacturer's protocol. Consequently, expression levels of selected genes were analyzed with qPCR. Five hundred nanograms of RNA were reverse transcribed to cDNA using reverse transcription reagents (Go Script, Promega, Madison, WI, USA) per the manufacturer's protocol. The total volume of 0.8 μl of cDNA was used per 20 μl real-time qPCR reaction using Power Up Sybr Green Master Mix (Applied Biosystem, Foster City, CA, USA) and run for 40 cycles in triplicates on a 7500 Fast Real Time PCR System machine (Applied Biosystem, Foster City, CA, USA), according to the manufacturer's instructions. Gene-specific primers are listed in Supplementary Table 1. Relative gene expression, normalized to beta-actin (ΔCt), and control GFP + VFEC or GFP− VFF (ΔΔCt) were calculated as fold change using the 2(-ΔΔCt) method[58]. If undetected, a cycle number 40 was assigned to allow fold change calculation. The data are presented as the average of the three biological replicates ± s.e.m.

The number of VFEC GFP + and VFF GFP− obtained by cell sorting was assessed to investigate the cell loss caused by the 5% CSE exposure. Flow data from three different differentiation rounds were analyzed. Three control VFEC GFP + samples (n = 3) and VFF GFP− samples (n = 3) were compared with three CSE exposed VFEC GFP + (n = 3) and VFF GFP− (n = 3). Statistical analysis was performed using the Student's t test. The results are reported as mean ± s.d., and were considered statistically significant at $p \leq 0.05$. The incomplete collagen gel digestion was likely to cause a variation in the size of an error bar in a VFF GFP− CSE exposed group, which reduced the number of VFF GFP− generated by the flow cytometry.

**Accession code**. RNA sequencing data for human primary vocal fold epithelial cell and human induced pluripotent stem cell-derived vocal fold basal progenitors have been deposited in the GEO database under accession code: GSE135548.

## Data availability
The authors declare that all data supporting the findings of this study are available within the article and its supplementary information files or from the corresponding author upon reasonable request. The source data underlying Figs. 1, 2, 3, 4, 10, and Supplementary Table 2 are provided as a Source Data file.

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

## Acknowledgements

This work was supported by grants NIH NIDCD R01 DC004336, R01 DC012773, and R01 DC014461. We gratefully acknowledge the University of Wisconsin–Madison Biotechnology Gene Expression Center & DNA Sequencing Facility for providing library preparation and next-generation sequencing services and flow cytometry facility at University of Wisconsin–Madison providing cell sorting for DE, AFE, and GFP-positive and GFP-negative cell populations. We also gratefully acknowledge Sierra Raglin for her expert assistance with the collagen construct sample preparation for this study. We thank Sean Palacek, PhD for technical advice.

## Author contributions

V.L. and S.L.T. designed the research; V.L. performed all experiments related to developmental derivation of human VF mucosa, including GFP transfection, characterization of hiPSC-derived cell populations, and functional experiments with 5% CSE exposure. X.C. isolated primary VF epithelial cells from human donors, passaged these cells, and isolated RNA for RNA sequencing; Z.W. and C.K. analyzed RNA data; and V.L. and S.L.T. wrote the paper. All authors approved the final paper.

## Additional information

**Competing interests:** The authors declare no competing interests.

