## [Peer Review File · Nature Communications]

Reviewers' Comments:

Reviewer #1:

Remarks to the Author:

The current manuscript has established human iPSC-derived vocal fold mucosa as a model for studying the impact of cigarette smoke (CSE)-induced inflammation on gene expression. The study uses cigarette smoke extract created as previously described for use in behavioral studies in rats. The authors state that 5% CSE is well tolerated by the in vivo cultures, as has been shown in the prior study. However, the prior study used a very different in vivo approach where there would have been much higher dilutions of CSE. Furthermore, that study did not examine the impact of CSE on cellular integrity. As a result, the authors need to directly quantitate cell numbers to show whether there is a selective loss of epithelial or fibroblast cells. Without such evidence, the observed changes in gene expression cannot be properly interpreted.

Reviewer #2:

Remarks to the Author:

Vlasta et al. present interesting findings that vocal fold epithelium can be derived from hiPS cells. They showed that these hiPSC-derived cells share similarity with human vocal epithelium in terms of gene expression and function. These hiPS-derived cells are able to form stratified epithelium with production of MUC4 and MUC1. Upon challenge with cigarette smoke extract the stratified epithelium formed by the hiPSC-derived cells produces inflammatory cytokines mimicking what have been observed in humans. These findings are significant in that this is the first report showing the successful derivation of vocal fold epithelium (VFE) from hiPSCs, which provides unlimited access to VFE. Some issues need to be considered.

It is quite novel that the authors generate VFE which forms stratified epithelium. Interestingly, in a recent paper Zhang et al. (Cell Stem Cell. 2018 Oct 4;23(4):516-529.) generate esophageal basal progenitor cells from hPSCs and these cells also form stratified epithelium. Since both VFE and esophageal cells are sequentially generated through endoderm/anterior foregut, it is suggested that authors consider to provide definitive evidence to distinguish their cells from esophageal epithelium. It is nice that the authors show MUC4 and MUC1 expression in 3D culture. How about gene expression by qPCR of 2D culture? And what is the expression level of these genes in their RNA seq data? In addition, Zhang et al. has published RNA seq of fetal and PSC-derived esophageal progenitor cells. Can the authors compare their sequencing data with these RNA datasets and identify VFE specific genes? It will be even better if they can further validate the expression of these genes in ipsc-derived VFE.

Minor issues:

1. In the trachea there is a new cell type called "hillock" cell which also expresses K13, k5 etc (Nature. 2018 Aug;560(7718):319-324), and Fgfs are essential for tracheal basal cell specification (Development. 2019 Feb 11;146(3)). Are these iPS-derived VFE expressing Nkx2.1? qPCR and immunostaining will help.
2. FoxE1 and Pax9 are expressed in the esophageal basal cells. Do vocal fold epithelia express these genes? If not these two genes could be good markers to distinguish VFE from esophageal lineage.
3. Ki67 expression seems different in Figure 6M and 8E although they are both from unperturbed VFE. Quantitative measurement of Ki67+ cells will be preferred for the cigarette smoke extract experiment.

Reviewer #3:

Remarks to the Author:

The manuscript describes method to differentiate iPS cells into vocal fold (VF) mucosa epithelial cells. First, authors generate iPS cells with GFP reporter, which is a useful technical approach. Next, they differentiate iPS cells into definitive endoderm (DE) and anterior foregut endoderm (AFE) using methods previously described. The next differentiation steps, when VF progenitor cells are obtained by applying various FGF concentrations and are further matured in 3D using air liquid interface culture, are a novel approach presented by the authors. There is a great need for reliable differentiation protocols that produce functional epithelial cells. Thus, if the protocol presented by the authors in this manuscript indeed yields functional epithelial cells and is highly reproducible, it will be of great benefit to the field. Last, authors show the utility of this in vitro model by applying 5% cigarette smoke and evaluating the response in both epithelial cells and fibroblasts.

My main concern and suggestion to the authors is to construct 3D air liquid interface tissues with human primary VFECs to compare the properties of the stratified epithelium from iPSC-derived VFECs and primary VFECs in 3D, as well as include native vocal fold mucosa tissue as a control in Figure 6. I believe that these controls are crucial for the evaluation of the properties and function of iPSC-derived cells.

Currently, I do not see the stratification of the iPSC-derived epithelial tissue in the H&E staining in Figure 6A or Figure 7A. Authors should include double staining of basal versus apical/suprabasal markers in constructed 3D tissues in order to confirm the stratification. Please, also include p63 staining. Authors should also perform double stain to show the separation of the epithelial tissue and lamina propria. For example, LAMA5 stain can be performed in conjunction with cytokeratin stains. Figure 9A which compares expression of the genes in 2D human primary VFECs and 3D human iPSC-derived VFECs is not a proper comparison, as 3D environment including but not limited to type I collagen, cell-cell interactions, growth factors present in the media, will impact the gene expression. In addition, in 3D as the epithelial cells stratify the tissue represents a heterogeneous population of cells (apical/suprabasal versus basal) with different gene expression signatures which is not seen in 2D cultures. Thus, it is imperative to compare 3D human primary VFECs and 3D human iPSC-derived VFECs.

Reproducibility of the presented differentiation protocol is a challenge. The authors should discuss how many times and how many tissues have been constructed using this protocol and should quantify any differences in the tissues between different rounds of differentiation, for example differences in the thickness of the epithelial tissue, difference in stratification, difference in the integrity of the basement membrane, difference in the number of Ki67 positive cells.

Currently, the main problem with iPSC-derived cells is their immaturity. The authors run into the same problem of VFPs being immature compared to 2D human primary VFECs and attempt to solve this challenge by culturing the progenitor cells in 3D. The maturity of the iPSC-derived cells remains unanswered. The authors show that with the treatment with 5% CSE the epithelial cells delaminate and migrate into the gel. Is this the response to 5% CSE treatment or is this the immaturity of iPSC-derived cells? 3D tissues constructed with primary VFECs and treated with 5% CSE are needed for the comparison. Also, in Figure 7 and 8 authors should show double staining with GFP and epithelial and fibroblasts markers to show that indeed iPSC-derived cells migrate into the type I collagen gel when treated with 5% CSE.

Authors state that cell sorting to separate VFECs and VFFs from 3D tissues resulted in cell death due to collagen gel digestion (Supplemental Figure S1 and S2), yet they also state that the selected 5% CSE was well tolerated by cells as they were viable (Line 276) which was determined by the same cell sorting analysis. Authors should clarify these statements.

For RNA-sequencing analysis, it is not clear how many technical and biological replicates were included in the analysis. Please, clarify.

Line 676. Please, clarify when the conditioned media from VFF cells was used. It is not clear when

it was used during the fabrication of 3D tissues.

Figure 1B. Please, provide a bright field image of the B10 colony in addition to the GFP.

Figure 2. Please, clarify what cells are stained in C-F, and H-K. Are they DE or AFE? In Line 138, authors state that proper derivation of DE and AFE was verified by ICC.

Figure 2. Please, clarify fold change over what is shown in B and G.

Figure 2 and Figure 3. Please, clarify how the levels of the transcripts compare between Figure 2G and Figure 3B.

Figure 6A. Please, provide a higher resolution H&E image. I am not seeing the VFF in this H&E stain. I am wondering if the VFF are present.

Figure 6D. Please, clarify how the dashed line was produced.

Figure 6L. LAMA5 and not Lam 5.

Figure 7 and Figure 8 captions should be re-considered. Currently, they are the same and do not reflect the different data that is shown in the figures.

Figure 9A. Please, clarify how human primary VFECs and iPSC-derived VFECs were cultured. Were they cultured in 2D or 3D?

Response to reviewers

Reviewer #1: The current manuscript has established human iPSC-derived vocal fold mucosa as a model for studying the impact of cigarette smoke (CSE)-induced inflammation on gene expression. The study uses cigarette smoke extract created as previously described for use in behavioral studies in rats. The authors state that 5% CSE is well tolerated by the in vivo cultures, as has been shown in the prior study. However, the prior study used a very different in vivo approach where there would have been much higher dilutions of CSE. Furthermore, that study did not examine the impact of CSE on cellular integrity. As a result, the authors need to directly quantitate cell numbers to show whether there is a selective loss of epithelial or fibroblast cells. Without such evidence, the observed changes in gene expression cannot be properly interpreted.

Response: We would like to thank the reviewer for their suggestions. We have directly quantified cells. Further, although we used a higher concentration of CSE than in the previous study, VFEC and VFF in the collagen matrix tolerated this level of CSE. We updated the results and provided further evidence to support our claims:

- Both cell types (VFEC and VFF) proliferated as confirmed by anti-Ki67 IF staining in control and CSE treated VF mucosae (Fig. 12A-C) and by the quantitative evaluation of the cell proliferation in VFEC and VFF in control and CSE treated cells (Fig. 12D).
- VFEC responded to the CSE by upregulating K14, K13, MUC1 and MUC4 and triggered VF epithelial remodeling as confirmed by the transcript analysis of cells isolated from constructs in control and CSE treated mucosae (Fig. 13 A) and by IF staining (Fig. 10 and 11).
- Exposure to CSE did not lead to a significant cell loss. We added quantitative assessment of the cell numbers obtained by the flow cytometry (Fig. 12E) as suggested by the reviewer. Our data show that there are no significant differences between the numbers of VFEC GFP+ and VFF GFP- in control and CSE treated cells. Notably, this method could be reliably used to separate the cell populations and study inflammatory cell responses to the CSE in both cells types simultaneously in future investigations (Fig.13C).

Reviewer #2: Vlasta et al. present interesting findings that vocal fold epithelium can be derived from hiPS cells. They showed that these hiPSC-derived cells share similarity with human vocal epithelium in terms of gene expression and function. These hiPS-derived cells are able to form stratified epithelium with production of MUC4 and MUC1. Upon challenge with cigarette smoke extract the stratified epithelium formed by the hiPSC-derived cells produces inflammatory cytokines mimicking what have been observed in humans. These findings are significant in that this is the first report showing the successful derivation of vocal fold epithelium (VFE) from hiPSCs, which provides unlimited access to VFE. Some issues need to be considered.

It is quite novel that the authors generate VFE which forms stratified epithelium. Interestingly, in a recent paper Zhang et al. (Cell Stem Cell. 2018 Oct 4;23(4):516-529.) generate esophageal basal progenitor cells from hPSCs and these cells also form stratified epithelium. Since both VFE and esophageal cells are sequentially generated through endoderm/anterior foregut, it is suggested that authors consider to provide

definitive evidence to distinguish their cells from esophageal epithelium. It is nice that the authors show MUC4 and MUC1 expression in 3D culture. How about gene expression by qPCR of 2D culture? And what is the expression level of these genes in their RNA seq data? In addition, Zhang et al. has published RNA seq of fetal and PSC-derived esophageal progenitor cells. Can the authors compare their sequencing data with these RNA datasets and identify VFE specific genes? It will be even better if they can further validate the expression of these genes in ipsc-derived VFE.

Response: We would like to thank the reviewer for these suggestions. As evidence that further supports that the VFE are not esophageal epithelium strengthens this investigation. In our revised manuscript we present new data that supports the fact that DE-AFE derived stratified epithelium are VFE and are indeed different from esophageal epithelium.

- First, we took advantage of publicly available RNAseq datasets of human fetal esophagus (Zhang et al 2018) and compared gene expression levels in human fetal esophageal epithelia with our RNA datasets. Our result demonstrate that out of 18842 extracted genes we found 8843 differentially expressed genes with the fold change value larger than 2 or less than ½ between hiPSC-derived VBP and human fetal esophageal epithelia. This is 1967 differentially genes more than differentially genes generated by the comparison between hiPSC-derived VBP and human primary adult VFEC. Result Section, page 11.
- We further selected 10 enriched genes upregulated in the VBP and human adult VFEC and 10 enriched genes upregulated in the fetal esophagus to identify potential tissue specific markers (Table 1). Notably, upregulated genes in the adult VF epithelium as well as hiPS-derived VBP include among the others FOXC2, KLF16, CDH13, CANX, and CARL. On the other hand, in the esophagus the upregulated genes include SOX2, SOX3, MUC22 and CDH6. Result Section, page 11.
- SOX2 gene was selected for validation of fully differentiated hiPSC-derived VFEC and native VF mucosa (Fig. 7E, F) along with a suggested marker FOXE1 (Fig. 7C, D). HiPSC-derived VBP and native VF mucosa were negative for both markers. Unfortunately, failure in RNA sequencing of FOXE1 gene in our dataset prevented us from including FOXE1 in the RNA seq data comparison.

We included new data comparing levels of MUC1 and MUC4 during the course of differentiation (Fig. 5B). Both mucins are significantly upregulated in fully differentiated hiPSC-VFEC to form the epithelial protective coat after 32 days in culture. Mucin expression levels are elevated in hiPSC-derived VFEC as compared to MUC1 and MUC4 expression levels in our RNA-seq dataset in human primary VFEC. This difference in mucin expression can be caused by the fact that in human native VF mucosa both mucins are expressed in suprabasal cell layers as confirmed by IF staining (Fig. 9 G -L). While 2D culture systems of human primary cells are preferentially composed of cells from the basal cellular compartment.

Minor issues:

1. In the trachea there is a new cell type called “hillock” cell which also expresses K13, K5 etc. (Nature. 2018 Aug;560(7718):319-324), and Fgfs are essential for tracheal basal cell specification (Development. 2019 Feb 11;146(3)). Are these iPS-derived VFE expressing Nkx2.1? qPCR and immunostaining will help.

- We performed NKX2-1 staining in hiPSC-derived VFEC and human native VF mucosa (Fig. 7A, B) showing that hiPSC-derived VFEC and human VF epithelium are negative for NKX2-1.

2. FoxE1 and Pax9 are expressed in the esophageal basal cells. Do vocal fold epithelia express these genes? If not these two genes could be good markers to distinguish VFE from esophageal lineage.

- Pax9 gene is highly expressed in the human VF epithelium as well as in hiPSC-derived VFEC. On the other hand, FOXE1 is neither expressed in the human VF mucosa nor hiPSC-derived VFEC (Fig. 7C, D). This gene was used to validate the specificity of VF epithelial differentiation along with SOX2 (Fig. 7E, F).

3. Ki67 expression seems different in Figure 6M and 8E although they are both from unperturbed VFE. Quantitative measurement of Ki67+ cells will be preferred for the cigarette smoke extract experiment.

- We added new staining and quantitative measurement of the cell proliferation (Ki67+ cells) in VFE and VFF in control and CSE exposed VF mucosae (Fig. 12A - D).

Reviewer #3: The manuscript describes method to differentiate iPS cells into vocal fold (VF) mucosa epithelial cells. First, authors generate iPS cells with GFP reporter, which is a useful technical approach. Next, they differentiate iPS cells into definitive endoderm (DE) and anterior foregut endoderm (AFE) using methods previously described. The next differentiation steps, when VF progenitor cells are obtained by applying various FGF concentrations and are further matured in 3D using air liquid interface culture, are a novel approach presented by the authors. There is a great need for reliable differentiation protocols that produce functional epithelial cells. Thus, if the protocol presented by the authors in this manuscript indeed yields functional epithelial cells and is highly reproducible, it will be of great benefit to the field. Last, authors show the utility of this in vitro model by applying 5% cigarette smoke and evaluating the response in both epithelial cells and fibroblasts.

My main concern and suggestion to the authors is to construct 3D air liquid interface tissues with human primary VFECs to compare the properties of the stratified epithelium from iPSC-derived VFECs and primary VFECs in 3D, as well as include native vocal fold mucosa tissue as a control in Figure 6. I believe that these controls are crucial for the evaluation of the properties and function of iPSC-derived cells.

Response: We would like to thank this reviewer for their careful and thoughtful comments. We have included new data and figures into the manuscript and compared expression of key morphological and functional genes of hiPSC-derived VFEC to native human VF mucosa (Figs. 8 and 9). Native VF mucosa is currently the only system that we could be used to validate the specificity of VF epithelial differentiation. Unfortunately, there are only a limited number of primary cells available which have limited growth potential whereby the reason we pursued developing iPSC-derived epithelial cells originally. Nevertheless, we attempted to cultivate a 3D construct from human primary epithelial cells. We were not able to re-establish stratification and other crucial epithelial characteristics such as cell adherent junctions or a well-developed basal

cell layer (please see Figure 1 here in the response to reviewers). Fortuitously, using native VF mucosa for validation of the hiPSC-derived cells provided complete expression patterning of mucins and genes expressed in the basal cellular compartment, such as double staining for p63 and cytokeratins K14, K13 and LAMA5, that future studies can use for the comparison.

Figure 1: Inadequate 3D Constructs from primary VFE.

Currently, I do not see the stratification of the iPSC-derived epithelial tissue in the H&E staining in Figure 6A or Figure 7A. Authors should include double staining of basal versus apical/suprabasal markers in constructed 3D tissues in order to confirm the stratification. Please, also include p63 staining. Authors should also perform double stain to show the separation of the epithelial tissue and lamina propria. For example, LAMA5 stain can be performed in conjunction with cytokeratin stains.

- To confirm the stratification, we added double staining of basal cellular marker p63+ and K14 to visualize basal cellular compartments and p63+ and K13 to visualize basal vs suprabasal cell layers in our engineered VF mucosae (Fig. 8A - L). We also added double staining of LAMA5 and p63 to show the separation of the epithelial tissue and

lamina propria (Fig. 9A – C) as suggested by the reviewer. The presence and pattern of these genes were validated using human native VF mucosa.

Figure 9A which compares expression of the genes in 2D human primary VFECs and 3D human iPSC-derived VFECs is not a proper comparison, as 3D environment including but not limited to type I collagen, cell-cell interactions, growth factors present in the media, will impact the gene expression. In addition, in 3D as the epithelial cells stratify the tissue represents a heterogeneous population of cells (apical/suprabasal versus basal) with different gene expression signatures which is not seen in 2D cultures. Thus, it is imperative to compare 3D human primary VFECs and 3D human iPSC-derived VFECs.

- We removed Figure 9A from results and compared expression of key morphological and functional VF epithelial genes in hiPSC-derived VFEC with human native VF mucosa by IF staining (Figs. 8 and 9).

Reproducibility of the presented differentiation protocol is a challenge. The authors should discuss how many times and how many tissues have been constructed using this protocol and should quantify any differences in the tissues between different rounds of differentiation, for example differences in the thickness of the epithelial tissue, difference in stratification, difference in the integrity of the basement membrane, difference in the number of Ki67 positive cells.

- Thank you for this suggestion. We had added information into both our Material and Methods and Discussion to address this concern.

Currently, the main problem with iPSC-derived cells is their immaturity. The authors run into the same problem of VFPs being immature compared to 2D human primary VFECs and attempt to solve this challenge by culturing the progenitor cells in 3D. The maturity of the iPSC-derived cells remains unanswered.

- We added the transcript analysis of hiPSC-derived VFEC after 32 days of differentiation and compared the expression levels of mucins, MUC 1 and MUC 4 and stratified markers p63, K14, K13 and K5, vs a simple epithelial marker K8 with earlier stages of differentiation DE (Day 4), AFE (Day 8) and VBP (Day 10 – during replating of cells on the collagen gel) (Fig. 5B). Robust increases in expression of stratified markers and upregulated expression levels of mucins suggest that reseeded cells on collagen gel can lead to maturation VBP into functional VF stratified epithelium.

The authors show that with the treatment with 5% CSE the epithelial cells delaminate and migrate into the gel. Is this the response to 5% CSE treatment or is this the immaturity of iPSC-derived cells? 3D tissues constructed with primary VFECs and treated with 5% CSE are needed for the comparison. Also, in Figure 7 and 8 authors should show double staining with GFP and epithelial and fibroblasts markers to show that indeed iPSC-derived cells migrate into the type I collagen gel when treated with 5% CSE.

- We performed the double staining with GFP and K14 in control vs CSE treated VF mucosae (Fig. 10 I - L). We did not see the GFP positive cells migrating into the collagen gel. We removed this statement from the Result and Discussion Sections.

Authors state that cell sorting to separate VFECs and VFFs from 3D tissues resulted in cell death due to collagen gel digestion (Supplemental Figure S1 and S2), yet they also state that the selected 5% CSE was well tolerated by cells as they were viable (Line 276) which was determined by the same cell sorting analysis. Authors should clarify these statements.

We rephrased these statements and updated the results providing additional evidence that hiPSC-derived VFEC as well as VFF in the collagen matrix tolerate the 5% CSE concentration.

- First, both cell types proliferate as confirmed by anti-Ki67 staining in control and CSE treated VF mucosae (Fig. 12A - C) and by quantitative evaluation of the cell proliferation in VFEC and VFF in control and CSE treated cells (Fig. 12D).
- Second, VFEC are capable of responding to CSE by upregulating the expression K14 K13, MUC1 and MUC4 and initiate VF epithelial remodeling as confirmed by the transcript analysis of cells isolated from constructs in control and CSE treated mucosae (Fig. 13 A, B) and by IF staining (Fig. 10 and 11).
- Third, exposure to CSE did not lead to a significant cell loss. We added quantitative assessment of the cell numbers obtained by the flow cytometry (Fig. 12E). Our data show that there are no significant differences between the numbers of VFEC GFP+ and VFF GFP- in control and CSE treated cells. This method could be reliably used to separate the cell populations and study inflammatory cell responses to the CSE in both cells types simultaneously (Fig.13C).

For RNA-sequencing analysis, it is not clear how many technical and biological replicates were included in the analysis. Please, clarify.

- We performed RNAseq analysis in two biological and two technical replicates – 4 samples total for each condition (human primary VFEC were isolated from two human donors and VBP were isolated from two different rounds of differentiation). We have updated this in the Materials and Methods.

Line 676. Please, clarify when the conditioned media from VFF cells was used. It is not clear when it was used during the fabrication of 3D tissues.

- We updated our schematic illustration of our differentiation protocol and added the number of days cells were exposed to conditional FAD medium (Fig. 5A). We also updated the Material and Methods and Result sections.

Figure 1B. Please, provide a bright field image of the B10 colony in addition to the GFP

- We provided a bright field image of the B10 colony as well as the bright field images of DE, AFE and VBP during differentiation (Fig. 1C, G, J and M).

Figure 2. Please, clarify what cells are stained in C-F, and H-K. Are they DE or AFE? In Line 138, authors state that proper derivation of DE and AFE was verified by ICC.

- We updated Figure 2 and added ICC data for DE differentiation as well (Fig. 2). DE ICC data are labelled C – F, while AFE ICC data are labelled H – K.

Figure 2. Please, clarify fold change over what is shown in B and G.

- We have clarified fold change in the Figure Legends. In both graphs B and G, the controls are hiPS cells. In graph B we used a log scale because of the values obtained; a value of 0 = 1 in control cells. In graph G, we were did not have to use a log scale and in this case a value of zero is zero expression.

Figure 2 and Figure 3. Please, clarify how the levels of the transcripts compare between Figure 2G and Figure 3B.

- Figure 3 was included in the results to show that modulation of FGF signaling pathway can trigger expression of stratified markers. We compared the transcript levels in experimental groups with AFE controls. To compare the changes in expression of stratified markers during the whole course of differentiation through DE, AFE and VBP up to VFEC we added new data (Fig. 5B). These data demonstrate that even if FGF signaling was capable of inducing expression of stratified markers in VBP, replating of VBP and their further differentiation on collagen gel leads to a robust upregulation of stratified markers.

Figure 6A. Please, provide a higher resolution H&E image. I am not seeing the VFF in this H&E stain. I am wondering if the VFF are present.

- We provided a higher resolution H&E image with hiPSC-derived VF epithelium and VFF in the collagen matrix (Fig. 8A)

Figure 6D. Please, clarify how the dashed line was produced.

- The Figure 6D was removed from the study and replaced with updated and more relevant images.

Figure 6L. LAMA5 and not Lam 5.

- We replaced LAM5 with LAMA5

Figure 7 and Figure 8 captions should be re-considered. Currently, they are the same and do not reflect the different data that is shown in the figures.

- These two figures were replaced with new data. Figure caption were, therefore, updated as well to reflect the data that are shown in these figures as suggested by the reviewer.

Figure 9A. Please, clarify how human primary VFECs and iPSC-derived VFECs were cultured. Were they cultured in 2D or 3D?

- Because of additional work completed for the revision and in response to all the reviewers, Figure 9 from our initial submission has been removed. In the Material and Methods we have noted that the primary VFECs were cultured in 2D and hiPSC-derived VFEC were cultured following protocols described in Figure 5A.

Reviewers' Comments:

Reviewer #1:

Remarks to the Author:

The current manuscript has established human iPSC-derived vocal fold mucosa as a model for studying the impact of cigarette smoke extract-induced inflammation on gene expression. The authors have responded well to the prior reviews and have provided evidence that cigarette smoke extract did not selectively reduce epithelial or fibroblast cells. This study not only establishes a new in vitro platform for examining exogenous insults to vocal fold mucosa, but also demonstrates the toxicity of tobacco smoke to these cells.

Frances Leslie

Reviewer #2:

Remarks to the Author:

The authors have addressed all the concerns raised by this reviewer. The new data are informative and significantly strengthen the manuscript.

Reviewer #3:

Remarks to the Author:

I would like to thank the authors for carefully addressing my previous comments and the comments of other reviewers. I believe that including human native VF mucosa as a control as well as double staining with relevant markers improved the manuscript. Authors also added relevant explanation about the generation of 3D tissues from primary VF epithelial cells in the response to reviewers. I have several minor comments about formatting issues and about newly added data in the revised manuscript.

Minor comments:

1. In Fig. 1D, I would keep labels of all the bands and keep the explanation about the iPSC clones in the Materials and Methods section. This was present in the original manuscript but was removed in the revised manuscript.
2. In Fig. 1D, it should be 3.8kbp and not 3.8kpb.
3. Authors changed scale bars between original and revised manuscript. Please, make sure and confirm that the scale bars are correct and consistent throughout the manuscript.
4. Authors revised the differentiation protocol that was present in the original manuscript. First, the AFE were derived for 5 days (original manuscript) and 4 days (revised manuscript). Next, EGF that was present in the media in the original methods has been removed in the revised manuscript. Next, the timing of the generation of 3D tissues has been changed between original and revised manuscript. Specifically, in the original manuscript AFE were treated with high concentration of FGFs in DMEM basal medium supplemented with EGF for 4 days, and then VBP were reseeded on collagen gels. IN the revised manuscript AFE were cultured for 2 days in 2D and then were seeded on collagen. Next, the duration of ALI culture is different between two versions of the manuscript. Please, clarify and make sure that the protocol is easy to follow and is reproducible.
5. In Fig. 2C, please include Sox17 and DAPI stained image.
6. In Fig. 3, please clarify how AFE controls were obtained. This information was present in the original manuscript and it is missing now. It is not clear how AFE controls are different from no FGF2 controls.
7. In Line 202, it should be Fig. 3B and not Fig 3B-G.
8. Please, clarify in method what VBP were used for RNA-seq. Are these the cells that were

cultured in the mix of FGFs? For how many days?

9. Line 239 and Table 1, is it Sox3 or Sox13?

10. What were the samples in Fig. 5B D32? Are they only VFECs flow sorted based on GFP expression or all the cells at day32?

11. In Fig. 7, it would be nice to have positive controls for NKX2.1 and FOXE1 staining. There is positive staining for Sox2 in previous figures but not for NKX2.1 and FOXE1.

11. Please, include in the methods section how KI-67 quantification was performed (Fig. 12D).

Please, comment on the big error bars in Fig. 12D and Fig. 12E.

12. Line 779, remove "anti-rabbit" because other secondaries were also used.

13. In Fig. 9, iPSC-derived and not hiPs-derived.

14. Fig. 13A (revised manuscript) and Fig. 9B (original manuscript) show different results except MUC1 and MUC4 that show consistent results. Originally, there was downregulation of K14 due to 5% CSE and now there is high upregulation in the levels of K14 due to 5% CSE. Please, comment on these differences.

Response to Reviewers

Reviewer 3

In Fig. 1D, I would keep labels of all the bands and keep the explanation about the iPSC clones in the Materials and Methods section. This was present in the original manuscript but was removed in the revised manuscript.

Response: Thank you for this suggestion. We have returned the explanation of hiPSC clones to the Material and Methods.

In Fig. 1D, it should be 3.8kbp and not 3.8kpb.

Response: We have correct this as suggested.

Authors changed scale bars between original and revised manuscript. Please, make sure and confirm that the scale bars are correct and consistent throughout the manuscript.

Response: We would like to apologize for the confusion. We have confirmed that the scale bars in all figures are correct and consistent.

Authors revised the differentiation protocol that was present in the original manuscript. First, the AFE were derived for 5 days (original manuscript) and 4 days (revised manuscript). Next, EGF that was present in the media in the original methods has been removed in the revised manuscript. Next, the timing of the generation of 3D tissues has been changed between original and revised manuscript. Specifically, in the original manuscript AFE were treated with high concentration of FGFs in DMEM basal medium supplemented with EGF for 4 days, and then VBP were reseeded on collagen gels. IN the revised manuscript AFE were cultured for 2 days in 2D and then were seeded on collagen. Next, the duration of ALI culture is different between two versions of the manuscript. Please, clarify and make sure that the protocol is easy to follow and is reproducible.

Response: We apologize for this confusion. We have optimized our protocol and document this in the manuscript which is easy to follow and reproducible.

In Fig. 2C, please include Sox17 and DAPI stained image.

Response: Thank you for the suggestion. We have included the Sox17 DAPI stained image.

In Fig. 3, please clarify how AFE controls were obtained. This information was present in the original manuscript and it is missing now. It is not clear how AFE controls are different from no FGF2 controls.

Response: AFE controls were collected at Day 8 of differentiation, while no FGFs samples were collected at Day 12 of differentiation along with other experimental groups. We apologize for removing this and have added this information back into the Result and Method sections.

In Line 202, it should be Fig. 3B and not Fig 3B-G.

Response: We apologize for this oversight. We have corrected this mistake.

Please, clarify in method what VBP were used for RNA-seq. Are these the cells that were cultured in the mix of FGFs? For how many days?

Response: VBP collected for RNA-seq were cultured in the mixture of FGFs for 4 days. This information has been added to the Result and Method sections.

Line 239 and Table 1, is it Sox3 or Sox13?

Response: We thank the reviewer for changing this error. We have corrected this to read Sox 13.

What were the samples in Fig. 5B D32? Are they only VFECs flow sorted based on GFP expression or all the cells at day32?

Response: D32 cells in Fig.5B are all cells collected by the digestion of the collagen gel. We have clarified this in the legend and manuscript.

In Fig. 7, it would be nice to have positive controls for NKX2.1 and FOXE1 staining. There is positive staining for Sox2 in previous figures but not for NKX2.1 and FOXE1.

Response: We have included positive controls. Human lungs were used as a positive control for NKX2.1; human tonsils were used as a positive control for FOXE1 (Fig. 5 I, L).

Please, include in the methods section how KI-67 quantification was performed (Fig. 12D). Please, comment on the big error bars in Fig. 12D and Fig. 12E.

Response: We added the how Ki67 quantification was performed into the Methods section. In order to comment on the big error bars in 12D (current Fig. 10A), we reviewed the original raw data to search for possible mistakes in calculations. Indeed, in VFF control and CSE exposed groups we wrongly calculated the percentage of Ki67 positive cells (11 out of 622 control VFF were positive for Ki67, which is 1.77% not 17.7% as stated; similarly, 6 out 324 CSE exposed VFF were positive for Ki67, which is 1.85% not 18.5% as stated). Specific to error bars in the Fig. 12E (current Fig. 10B), the number of cells generated by the flow cytometry is influenced by the digestion of the collagen gel and cell dissociation. Only single cells that pass through the sorter are calculated. Improper digestion of collagen gel can lead to decreased number of VFF obtained by the cell sorting. These causes larger variability in our results.

Line 779, remove “anti-rabbit” because other secondaries were also used.

Response: We have removed “anti-rabbit” as suggested.

In Fig. 9, hiPSC-derived and not hiPs-derived.

Response: Thank you for catching this error. We have corrected this error.

Fig. 13A (revised manuscript) and Fig. 9B (original manuscript) show different results except MUC1 and MUC4 that show consistent results. Originally, there was downregulation of K14 due to 5% CSE and now there is high upregulation in the levels of K14 due to 5% CSE. Please, comment on these differences.

Response: In the original manuscript, cells obtained from the whole constructs were first sorted based on GFP expression and then analyzed with qPCR. During cell sorting many cells can be lost due to incomplete gel digestion or cell dissociation, which affects the expression of structural genes. This was ineffective. During the period that the paper was in review, we optimized this process. In the revised manuscript, we digested the collagen gel and collected all cells from the whole constructs. This improved our RNA extraction and allowed for precise measurements. The change in K14 expression is secondary to having a better yield in RNA. Because K14 is only expressed in epithelial cells, even though RNA is collected from all cells, we can infer that the K14 is epithelial in origin.